# Impaired spatial memory codes in a mouse model of Rett syndrome

Sara E Kee[1,2], Xiang Mou[1], Huda Y Zoghbi[2,3,4,5,6,7]*, Daoyun Ji[1,2]*

[1]Department of Molecular and Cellular Biology, Baylor College of Medicine, Houston, United States; [2]Department of Neuroscience, Baylor College of Medicine, Houston, United States; [3]Department of Molecular and Human Genetics, Baylor College of Medicine, Houston, United States; [4]Department of Neurology, Baylor College of Medicine, Houston, United States; [5]Department of Pediatrics, Baylor College of Medicine, Houston, United States; [6]Howard Hughes Medical Institute, Baylor College of Medicine, Houston, United States; [7]Jan and Dan Duncan Neurological Research Institute, Texas Children's Hospital, Houston, United States

**Abstract** The *Mecp2+/-* mouse model recapitulates many phenotypes of patients with Rett syndrome (RTT), including learning and memory deficits. It is unknown, however, how the disease state alters memory circuit functions in vivo in RTT mice. Here we recorded from hippocampal place cells, which are thought to encode spatial memories, in freely moving RTT mice and littermate controls. We found that place cells in RTT mice are impaired in their experience-dependent increase of spatial information. This impairment is accompanied by an enhanced baseline firing synchrony of place cells within ripple oscillations during rest, which consequently occludes the increase in synchrony after a novel experience. Behaviorally, contextual memory is normal at short but not long time scale in RTT mice. Our results suggest that hypersynchrony interferes with memory consolidation and leads to impaired spatial memory codes in RTT mice, providing a possible circuit mechanism for memory deficits in Rett Syndrome.
DOI: https://doi.org/10.7554/eLife.31451.001

**\*For correspondence:**
hzoghbi@bcm.edu (HYZ);
dji@bcm.edu (DJ)

## Introduction

Rett syndrome, a postnatal neurodevelopmental disorder, is characterized by a period of normal development lasting until 6–18 months of life, followed by a period of regression and development of motor deficits, autonomic dysfunction, and intellectual disability (*Chahrour and Zoghbi, 2007*). Rett syndrome is caused by loss-of- function mutations in the X-linked gene *MECP2*, which encodes Methyl-CpG Binding Protein 2, a protein proposed to regulate chromatin states (*Amir et al., 1999*). Rett syndrome is primarily seen in heterozygous females (*Chahrour and Zoghbi, 2007*). Accordingly, female mice heterozygous for the deletion of *Mecp2* (*Mecp2+/-*; hereafter referred to as RTT mice) have been used to model the syndrome (*Guy et al., 2001*). RTT mice recapitulate many features of the human disorder, including the cognitive phenotypes (*Guy et al., 2001*). For example, RTT mice display deficits in contextual fear memory (*Samaco et al., 2013*) and spatial reference memory (*Hao et al., 2015*). However, how changes in the learning and memory neural circuits in RTT mice give rise to these deficits in vivo is not understood.

The hippocampus, a key area in the brain's learning and memory circuits, is critical for both contextual fear memory (*Phillips and LeDoux, 1992*; *Fanselow, 2000*) and spatial memory (*Morris et al., 1982*). In particular, spatial memory is thought to be encoded by hippocampal 'place cells', which fire when the animal enters one or a few locations (place fields) in an environment (*O'Keefe and Dostrovsky, 1971*; *Wilson and McNaughton, 1993*). Place field locations are mostly stable over time and become more refined as the animal's experience in the environment increases

(*Thompson and Best, 1990*; *Frank et al., 2004*; *Cacucci et al., 2007*). This experience-dependent refinement of spatial memory code is thought to involve memory consolidation in rest periods and sleep (*Diekelmann and Born, 2010*; *Walker and Stickgold, 2004*), during which place cells are activated synchronously within short (~100 ms) time windows of high-frequency (100–250 Hz) oscillations, called sharp-wave ripples, in the local-field potentials (LFPs) of the hippocampal CA1 area (*Buzsáki et al., 1992*; *Csicsvari et al., 2000*; *Buzsáki, 1989*; *Wilson and McNaughton, 1994*). The synchronous firing in ripples is believed to induce synaptic plasticity, which transforms short-term memories to long-term memories (*Buzsáki, 1989*; *Wilson and McNaughton, 1994*; *Tatsuno et al., 2007*; *Ji and Wilson, 2007*; *Sirota et al., 2003*). Consistent with this idea, disrupting ripples is shown to impair experience-dependent learning (*Girardeau et al., 2009*; *Ego-Stengel and Wilson, 2010*). Despite the importance of place cells and their activities during ripple oscillations in hippocampus-dependent learning and memory, how these cells contribute to cognitive deficits in animal models of neurodevelopmental disorders is unknown.

To understand the neural circuit basis of memory deficits in RTT mice, we set out to study how hippocampal place cell activities are altered in RTT mice. In particular, our previous study shows that hippocampal cells in RTT mice tend to display an exaggerated level of synchrony (hypersynchrony) in vitro and in vivo (*Lu et al., 2016*). We therefore focused on the impact of hypersynchrony on spatial memory codes in freely behaving RTT mice. Specifically, we set out to test the hypothesis that hypersynchrony interferes with ripple-dependent memory processing and consequently impairs the long-term refinement of spatial memory codes in RTT mice.

## Results

To test the hypothesis we recorded place cells and LFPs using tetrodes (*Cheng and Ji, 2013*), from the hippocampal CA1 of adult RTT mice and their wildtype (WT) littermates that were at least 3 months of age (*Figure 1A*). The recordings were made as animals ran back and forth (two trajectories) for food rewards on either a novel or familiar linear track, repeatedly for two sessions (Run 1, Run 2). There was also a rest session before (Pre-run) and after (Post-run) the first run session in a majority of the recordings (*Figure 1B*). Each of these run and rest sessions was 10–15 min. Animals (*N* = 10 RTT, 9 WT mice) were first trained to perform the task on the familiar track in a familiar room for at least 2 weeks before the recording on the same track began. Then, a subset of these animals (*N* = 9 RTT, 7 WT mice) was placed in a new room and recorded following the same procedure on a novel track, which had a similar shape to the familiar track. The animals had never been exposed to the novel room or the novel track before. During the recording on both the familiar and novel tracks, RTT and WT mice displayed similar performance, as measured by the number of times (laps) an animal traversed a trajectory in a run session and the animal's average speed of running (*Figure 1C,D*). We did not observe any seizures behaviorally and did not detect any epileptic-like activities in any of the recorded LFPs.

### Place cells in RTT mice contained less spatial information in the familiar, but not the novel, environment

We first analyzed place cell activities on the novel track. A total of 47 cells from 9 RTT mice (RTT cells) and 93 cells from 7 WT mice (WT cells) were recorded on the novel track. Of these, 39 RTT cells and 84 WT cells were classified as putative pyramidal cells (with mean firing rate <7 Hz; *Table 1*). We defined a run-active cell as a putative pyramidal cell active (with mean firing rate >0.5 Hz) on at least one trajectory during at least one run session. The percentage of run-active cells among all putative pyramidal cells in RTT mice (79%, *N* = 31) did not significantly differ from that in WT (63%, *N* = 53) mice (p=0.069, binomial test). The median firing rate of the run-active cells in RTT mice (median [25–75%] range: 1.8 [0.71 3.2] Hz, *N* = 31) was also not significantly different from that in WT mice (1.1 [0.57 2.3] Hz, *N* = 53; p=0.10, Wilcoxon ranksum test) on the novel track.

To examine place fields of run-active cells on the novel track, we linearized an animal's locations on a track trajectory. For each cell active on a trajectory in a run session, we plotted the cell's spike raster over the linearized locations during every running lap and computed a firing rate curve (average firing rates over all laps versus linearized locations of the trajectory). On the novel track, raster plots and rate curves show that RTT and WT cells fired spikes at specific locations (place fields) along a trajectory (*Figure 2A*). To quantify firing specificity, we computed a cell's spatial information

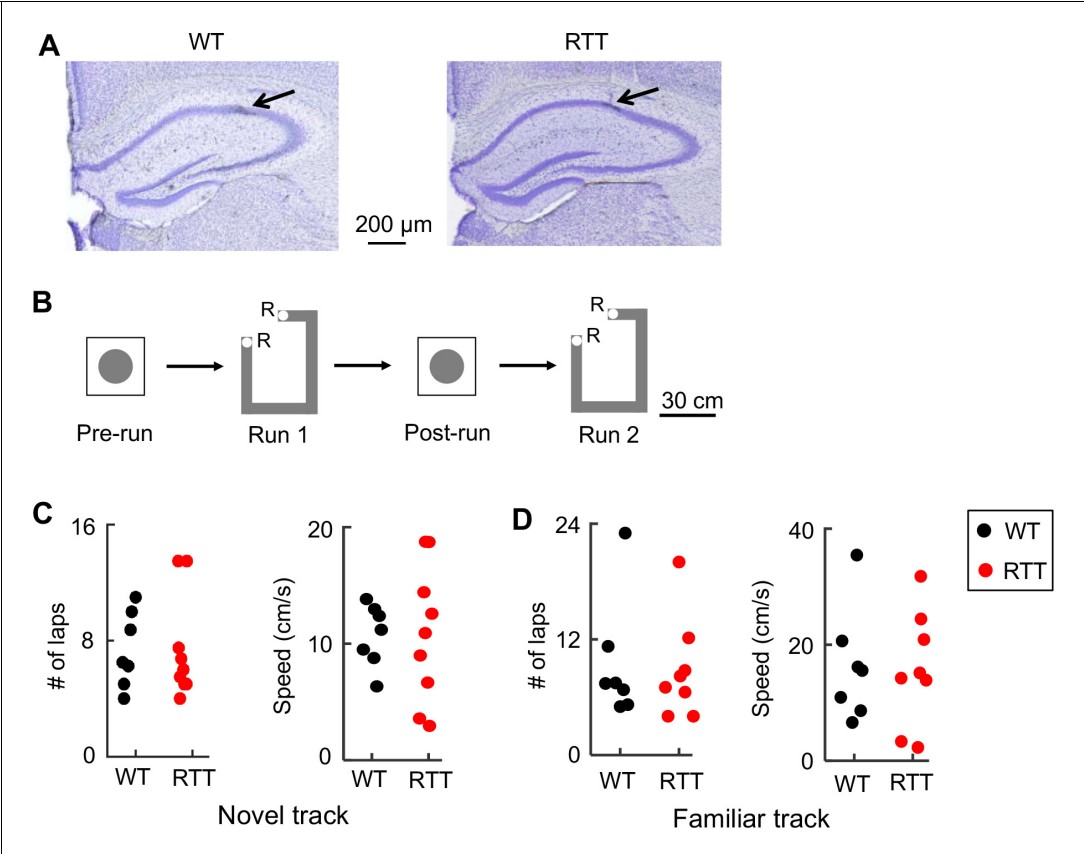

**Figure 1.** Recording of CA1 cells in WT and RTT mice during track-running and during rest. (**A**) Coronal brain sections of a WT and a RTT mouse stained with cresyl violet. *Arrow*: tetrode location. (**B**) Recording sessions and schematics of a track and a rest box. Animals ran back and forth on the track for food reward (R: reward sites) for two sessions (Run 1 and Run 2) and rested in the box before (Pre-run) and after (Post-run) the first run session. (**C**) Number of laps and running speed of WT and RTT mice on the novel track. Number of animals: N = 7 (WT), 9 (RTT). Wilcoxon ranksum tests: p=0.66 for number of laps; p=0.65 for speed. (**D**) Same as in (**C**), but on the familiar track. Number of animals: N = 7 (WT), 8 (RTT). Wilcoxon ranksum tests: p=0.93 for number of laps, p=0.96 for speed.

DOI: https://doi.org/10.7554/eLife.31451.002

(**Skaggs et al., 1993**), which measures, in bits per spike, the amount of information the cell's firing provides about the animal's location on a trajectory. We found that the spatial information of RTT cells was similar to that of WT cells (**Figure 2B**) on the novel track. We then quantified how consistent a cell's firing on a trajectory was between Run 1 and Run 2 by firing stability, which is the Pearson correlation between the rate curves of a cell in the two sessions (**Ciupek et al., 2015**). We found that the stability of RTT cells on the novel track was significantly reduced from that of WT cells (**Figure 2B**). We then identified individual place fields of each cell and determined whether place field properties were altered in RTT mice. We found a greater number of place fields per cell per active trajectory in RTT (median [25–75%] range: 2 [1 3]; N = 31 cells) than that in WT cells (1 [1 2], N = 53; p=0.049, Wilcoxon ranksum test), but the place field length in RTT mice was shorter (**Figure 2B**). The median peak firing rate within place fields was comparable between RTT and WT cells on the novel track (**Figure 2B**). These quantifications show that place cells in RTT mice contained similar information on the novel track as those in WT mice, but encoded space by more place fields with shorter lengths.

We next performed the same analyses on RTT and WT cells active on the familiar track (**Figure 2C**), where we recorded 171 cells from 8 RTT mice and 136 cells from 7 WT mice. Among these, 135 RTT cells and 102 WT cells were putative pyramidal cells (**Table 1**). The percentage of run-active cells among all putative pyramidal neurons in RTT mice (77%, N = 103) was greater than that in WT (64%, N = 65) mice (p=0.035). The median firing rate of these run-active neurons in RTT

**Table 1.** WT and RTT animals used in the experiment and number of putative pyramidal cells recorded from each animal on the novel and familiar track.
Putative interneurons were not included. x: no recording was made, -: behavioral and neural data not analyzed because of insufficient number of laps during running.

| Genotype | Novel track cells | | Familiar track cells | |
|---|---|---|---|---|
| | Putative pyramidal | Active | Putative pyramidal | Active |
| WT | 23 | 12 | 36 | 27 |
| WT | - | - | 1 | 1 |
| WT | 21 | 15 | 8 | 5 |
| WT | 8 | 6 | 6 | 4 |
| WT | 10 | 6 | 26 | 11 |
| WT | 1 | 1 | 7 | 6 |
| WT | 9 | 4 | 18 | 11 |
| WT | 12 | 9 | x | x |
| RTT | - | - | 36 | 29 |
| RTT | 5 | 5 | 5 | 4 |
| RTT | 6 | 5 | 8 | 6 |
| RTT | 6 | 5 | 34 | 27 |
| RTT | 7 | 3 | 19 | 14 |
| RTT | 10 | 8 | 30 | 23 |
| RTT | 1 | 1 | 3 | 0 |
| RTT | 1 | 1 | 0 | 0 |
| RTT | 0 | 0 | x | x |
| RTT | 3 | 3 | x | x |
| WT total | 84 | 53 | 102 | 65 |
| RTT total | 39 | 31 | 135 | 103 |

DOI: https://doi.org/10.7554/eLife.31451.003

(1.8 [1.1 3.0] Hz, $N$ = 103) mice was significantly increased from that in WT mice (1.1 [0.6 2.1] Hz, $N$ = 65; p=0.0012). However, the spatial information of run-active RTT cells was significantly reduced (by 63%), compared to that of WT run-active cells, while the stability was similar (**Figure 2D**). The number of place fields per cell was similar between RTT (2 [1 2], $N$ = 103 cells) and WT (1 [1 2], $N$ = 65) cells (p=0.16, Wilcoxon ranksum test) on the familiar track, but the place field length of RTT cells was significantly larger than that of WT cells (**Figure 2D**). The median peak firing rate within place fields was comparable between RTT and WT cells (**Figure 2D**). The results suggest that the reduced specificity of RTT cells on the familiar track was due to larger place fields.

To gain more understanding of this abnormality, we compared the firing specificity and place field properties of WT or RTT cells between the novel and familiar tracks. In WT mice, the median spatial information of place cells was greatly increased (300%) from the novel to familiar track (p=$3.3\times10^{-16}$, Wilcoxon ranksum test), indicating an experience-dependent refinement of firing specificity in WT cells. The number of place fields remained similar between the novel and familiar track, but the field length became significantly smaller (p=0.00056) and median peak firing rate within place fields was significantly increased (p=$1.5\times10^{-7}$). The result suggests that experience refines place cells by narrowing place field sizes and increasing firing rates. In RTT mice, the spatial information of RTT cells was only modestly increased between novel and familiar tracks (48%, p=0.008), indicating a reduction in experience-dependent refinement of firing specificity in RTT mice, compared to that in WT mice. The number of place fields was similar between the novel and familiar track and the peak rate increased (p=$4.5\times10^{-5}$), as in WT mice. However, place fields in RTT mice were relatively narrow on the novel track and became significantly widened on the familiar track (p=$2.4\times10^{-8}$). The comparison indicates that, although place cells in RTT mice still increased

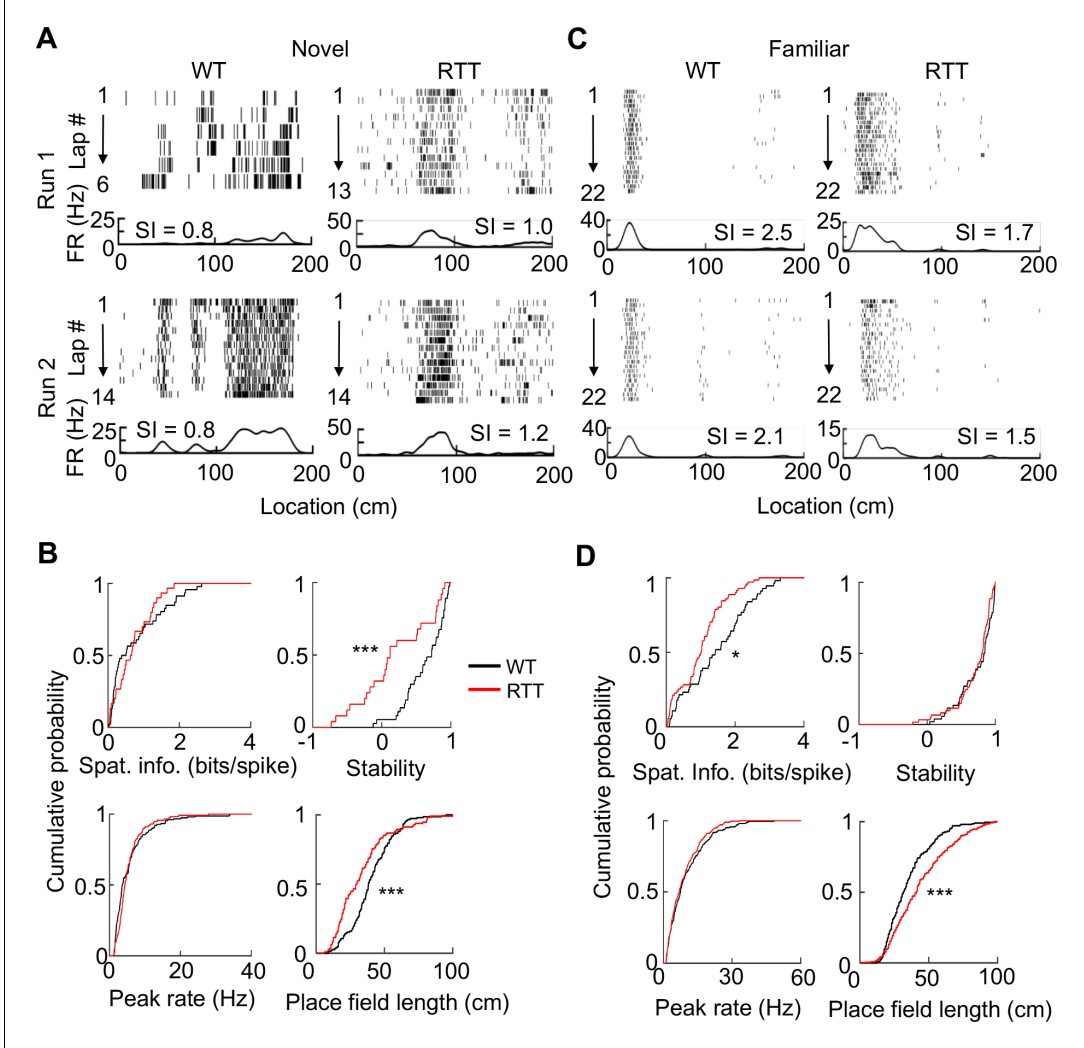

**Figure 2.** Place cells in RTT mice had less firing specificity in the familiar, but not in the novel, environment. (**A**) Firing activities of an example WT and RTT cell during running on the novel track in two sessions (Run 1, Run 2). For each panel, the top shows spike raster of a cell during every running lap on a linearized trajectory (running direction: from left to right). Each row is a lap and each tick is a spike. The bottom trace is the firing rate curve averaged over all laps in a session. (**B**) Distributions of spatial information, stability, peak firing rate within place fields, and place field length for WT and RTT place cells on the novel track. Spatial information: p=0.67, Wilcoxon ranksum test; number of cells by active trajectories: N = 79 (WT), 51 (RTT). Stability: ***p=0.0008; number of cells by active trajectory in both run sessions: N = 37 (WT), 25 (RTT). Peak rate: p=0.25; place field length: ***p=0.0001; number of place fields: N = 151 (WT), 115 (RTT). (**C**) Same as in (**A**), but for example cells on the familiar track. (**D**) Same as in (**B**), but for place cells on the familiar track. Spatial information: *p=0.016, number of cells by active trajectories: N = 87 (WT), 120 (RTT). Stability: p=0.33; number of cells by active trajectories in both run sessions: N = 52 (WT), 61 (RTT). Peak rate: p=0.26; place field length: ***p=$2.1 \times 10^{-8}$; number of place fields: N = 206 (WT), 280 (RTT).

DOI: https://doi.org/10.7554/eLife.31451.004

The following figure supplements are available for figure 2:

**Figure supplement 1.** Place cells in RTT mice were impaired in experience-dependent increase in spatial information, after cells active on the novel and familiar tracks were down-sampled.

DOI: https://doi.org/10.7554/eLife.31451.005

**Figure supplement 2.** Lap-by-lap spatial information of place cells on the novel and familiar tracks.

DOI: https://doi.org/10.7554/eLife.31451.006

spatial information with experience, likely due to higher peak firing rates, this increase was much less than that in WT mice mainly because of larger place fields.

We need to point out that there were fewer cells recorded on the novel track than on the familiar track. The smaller sample size could render a true effect involving the novel track statistically

insignificant. Therefore, we performed a downsampling analysis, in which we re-sampled down the number of cells active on the familiar track in both WT and RTT mice, as well as the number of WT cells active on the novel track, to the same number of RTT cells (31) on the novel track. We repeated the downsampling 200 iterations and examined a key effect involving place cells on the familiar track, spatial information, for each iteration. We found that 42% of the iterations produced a significant difference in spatial information between WT and RTT cells on the familiar track and 3% in spatial information on the novel track (*Figure 2—figure supplement 1*). The numbers suggest that, although the difference on the familiar track was more frequently observed than that on the novel track, the downsampling did reduce the statistical power for identifying a significant difference between WT and RTT cells on the familiar track. On the other hand, 98% of the iterations produced a significant difference in spatial information of WT cells between the novel and familiar tracks, but only 25% in RTT cells, suggesting that the experience-dependent refinement was robust in WT mice, but largely disappeared in RTT mice, with the downsampled cells. Therefore, although the downsampling led to less statistical power, the reduced experience-dependent refinement in RTT cells, compared to that in WT cells, was reproduced even with the downsampled cells.

We then examined the dynamic change in spatial information at a shorter time scale of within a recording day. We computed the lap-by-lap spatial information during both Run 1 and Run 2 on the novel and familiar track (*Figure 2—figure supplement 2*). On the novel track, WT and RTT cells showed similar spatial information in early and late laps during Run 1 and throughout Run 2. On the familiar track, WT cells started both Run 1 and Run 2 with already high spatial information in early laps, but RTT cells started with a low level of spatial information similar to that on the novel track. This suggests that place cells in RTT mice could produce specific firing from ongoing sensory experience on the track. However, the specificity was not maintained between running sessions or across days.

Finally, to understand whether place fields of RTT cells were impaired at a fine level, we examined theta phase precession, which establishes a fine correlation between place cell firing and location (*O'Keefe and Recce, 1993*). We found that theta phases of both RTT and WT place cells precessed as animals passed through their place fields (*Figure 3A*). We quantified theta phase precession by an optimal linear regression (*O'Keefe and Recce, 1993*) and a circular-linear correlation (*Ravassard et al., 2013*) between spike theta phases and positions within a place field. The linear correlation and the slope of the linear regression did not significantly differ between WT and RTT cells on either the novel or the familiar track (*Figure 3B*). However, there was a small, yet significant, reduction in the circular-linear correlation of RTT cells in the familiar environment (11%), compared to that of WT cells, but not in the novel environment (*Figure 3C*). The analysis suggests that the fine level of correlation between firing activity and location within place fields was present in RTT mice, but could be slightly weaker in the familiar environment, possibly related to their larger place fields.

## Place cell synchrony failed to increase after novel experience in RTT mice

Next, we examined ripples in the rest session before (Pre-run) and after (Post-run) the first spatial experience (Run 1) on a given day in either the novel or familiar room. For each animal and each rest session, we identified individual ripple events in LFPs (*Figure 4A*) and quantified ripple properties by occurrence rate (number of ripples per second), duration, amplitude, and frequency. Each of these parameters was averaged over all ripple events in a rest session. In the novel environment, ANOVA analysis shows that there was no significant effect of genotype or significant interaction between genotype and rest session in ripple rate, duration, amplitude, or frequency (*Figure 4B*). Similar findings were made in the familiar environment (*Figure 4C*). The data suggest that ripples in RTT and WT mice were qualitatively similar in rest sessions.

We then examined firing rates of CA1 cells within ripple events (*Figure 4—figure supplement 1*). In the novel room, the firing rates of all putative pyramidal cells or run-active cells were not significantly different between WT and RTT in either Pre- or Post-run. In the familiar room, the median firing rate of all putative pyramidal cells in RTT mice was similar to that in WT mice in both Pre-run and Post-run, but run-active cells had higher firing rate in RTT mice than in WT mice, especially in Pre-run. This firing rate increase suggests that run-active cells might fire in ripples more often in RTT mice than in WT mice in the familiar room. Indeed, run-active cells in RTT mice participated in ripples with a higher rate and fired more spikes per ripple event in the familiar room, but not in the novel

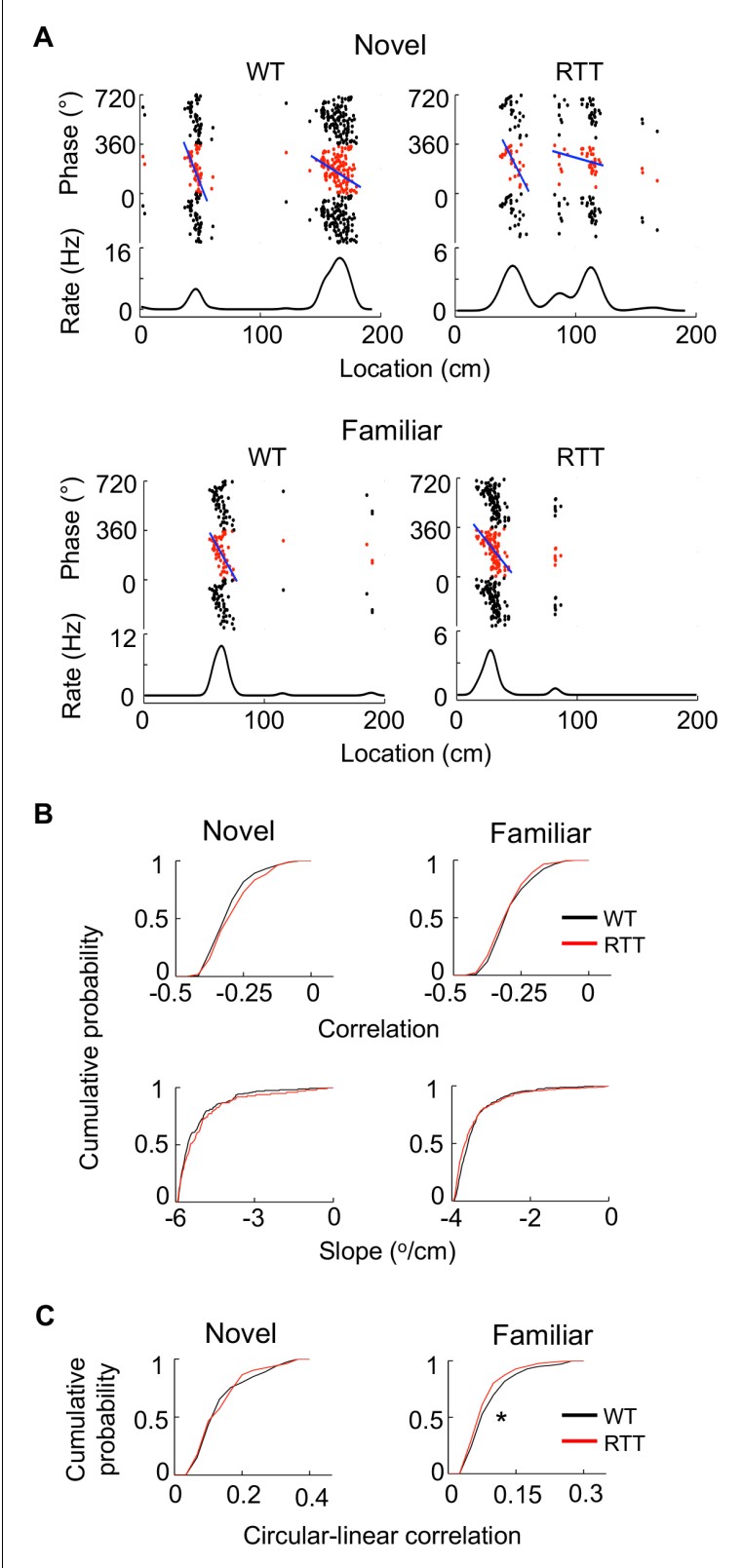

**Figure 3.** Place cells in RTT mice displayed largely normal phase precession on the novel and familiar track. (**A**) Example cells with theta phase precession from a WT and RTT animal on the novel and familiar track. For each panel, the top depicts theta phases of a cell versus positions on the track plotted three cycles and the bottom is the cell's mean firing rate. Solid line: linear regression of the theta phases versus positions. (**B**)

*Figure 3 continued on next page*

*Figure 3 continued*

Cumulative distributions of linear correlations and slopes of theta phase and position regressions for place fields in the WT and RTT mice on the novel and familiar tracks. Novel correlation: p=0.11, Wilcoxon ranksum test; slope: p=0.16; number of place fields: *N* = 160 (WT), 103 (RTT). Familiar correlation: p=0.15; slope: p=0.053; *N* = 237 (WT), 293 (RTT). (C) Cumulative distributions of circular-linear correlations of theta phase and position regressions for place fields in the WT and RTT mice on the novel and familiar tracks. Novel: p=0.77, Wilcoxon ranksum test; number of place fields: *N* = 160 (WT), 103 (RTT). Familiar: *p=0.016; *N* = 237 (WT), 293 (RTT).
DOI: https://doi.org/10.7554/eLife.31451.007

room (*Figure 4—figure supplement 2*). In contrast, run-inactive cells did not differ between WT and RTT mice in either of these parameters in either the novel or familiar room (*Figure 4—figure supplement 3*), providing a control that the difference in ripple participation between WT and RTT mice was not due to differences in resting behavior or ripple events per se.

It is known that place cells active in a new spatial experience tend to fire synchronously during ripples immediately after the experience more frequently than during ripples before (*Wilson and McNaughton, 1994*; *Skaggs and McNaughton, 1996*). We therefore asked whether this experience-dependent increase in firing synchrony was altered in RTT mice. We examined firing activities of run-active cells within ripples in Pre- and Post-run in the novel room. In WT mice, run-active cells rarely fired together during ripples in Pre-run, but did so more frequently in Post-run. However, in RTT mice run-active cells already fired together in Pre-run and they appeared not to do so more frequently in Post-run (*Figure 5A*). To quantify this, we computed a normalized pair-wise cross-correlogram (*Sirota et al., 2003*; *Ciupek et al., 2015*) within ripples in Pre-/Post-run for each pair of run-active cells that were also recorded in Pre-run and Post-run. Cross-correlograms of these cell pairs in WT mice showed low correlation around the time lag 0 in Pre-run and relatively high correlation in Post-run (*Figure 5B*). However, cross-correlograms in RTT mice showed high correlation around the time lag 0 already in Pre-run, without further increase in Post-run (*Figure 5B*). We used the average correlation value around time lag 0 (within [−50 50] ms) as a measure of 'coactivity' for each pair of cells (*Wilson and McNaughton, 1994*). We then compared the coactivity values of all run-active pairs between RTT and WT mice to evaluate their difference in firing synchrony at the cell population level. We found that the coactivity in WT mice was significantly increased from Pre- to Post-run (*Figure 5C*), indicating an experience-dependent increase in synchrony following running on the novel track, as expected (*Wilson and McNaughton, 1994*). In RTT mice, however, the coactivity was not significantly different between Pre-run and Post-run, indicating a lack of increase in synchrony after the novel track experience (*Figure 5C*). Moreover, the Pre-run coactivity was significantly higher in RTT mice than that in WT mice, whereas the Post-run coactivity was not significantly different. Therefore, run-active cells in RTT mice failed to increase their synchrony following a novel experience and this failure appeared to be due to their higher than normal synchrony (hypersynchrony) prior to the experience, consistent with our previous study (*Lu et al., 2016*).

The finding of hypersynchrony among run-active cells in Pre-run leads to the question of whether a general hypersynchrony within ripples existed among all putative pyramidal cells, unrelated to any track-running experience. We first re-analyzed the correlograms in Pre-run before the novel-track running, but for all pairs of putative pyramidal cells in RTT and WT mice. Indeed, we found that the coactivity in RTT mice was significantly higher than that in WT mice (*Figure 5—figure supplement 1*). Second, we analyzed the coactivity of run-active cells in the familiar environment. We found a similar hypersynchrony among run-active cells in RTT mice, relative to those in WT mice, within ripples in both Pre- and Post-run (*Figure 5—figure supplement 2*), although the coactivity of RTT cells in Post-run now became significantly higher than that in Pre-run, likely because the repetitive experience on the familiar track eventually further increased the cells' coactivity (see Discussion). Finally, we computed the coactivity of run-silent cells within ripples (Pre- and Post-run in novel and familiar environments combined) and found that the coactivity was higher in RTT mice than that in WT mice (*Figure 5—figure supplement 3*). The results suggest a general hypersynchrony among CA1 neurons in RTT mice.

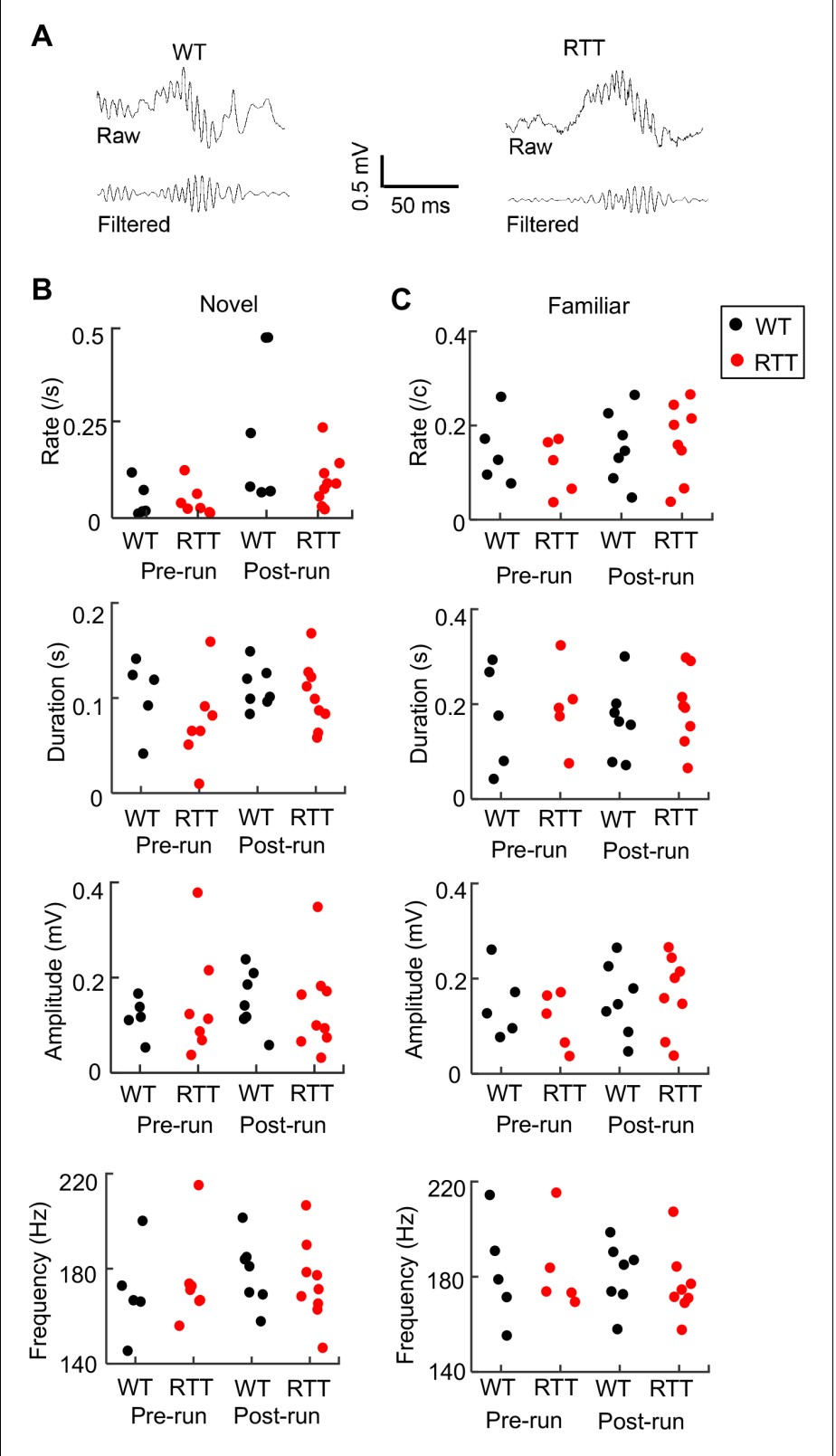

**Figure 4.** Ripple parameters were similar between WT and RTT mice during resting before (Pre-run) and after (Post-run) running on the novel and familiar track. (**A**) Raw and filtered (pass band: 100–250 Hz) LFP traces from a WT and RTT animal. (**B**) Ripple occurrence rate, duration, amplitude, and frequency during resting before (Pre-run) and after (Post-run) running on the novel track. Each dot is the parameter averaged over all ripple events in an

*Figure 4 continued on next page*

*Figure 4 continued*

animal. Number of animals: $N$ = 5 (WT), 7 (RTT) for Pre-run and 7 (WT), 9 (RTT) for Post-run. ANOVA analysis reveals no significant effect of genotype on occurrence rate ($F_{(1, 26)}$=2.0, p=0.17), duration ($F_{(1, 26)}$=1.8, p=0.19), amplitude ($F_{(1, 26)}$=0.04, p=0.84), or frequency ($F_{(1, 26)}$=0, p=1.0). There was a significant effect of session on occurrence rate ($F_{(1, 26)}$=6.8, *p=0.015), but not on duration ($F_{(1, 26)}$=1.6, p=0.22), amplitude ($F_{(1, 26)}$=0.14, p=0.71), or frequency ($F_{(1, 26)}$=0.33, p=0.57). There was no significant effect of the interaction between genotypes and resting sessions (Pre- and Post-run) on occurrence rate ($F_{(1, 24)}$=1.8, p=0.20), duration ($F_{(1, 24)}$=0.54, p=0.47), amplitude ($F_{(1, 24)}$=0.04 P=0.84), or frequency ($F_{(1, 24)}$=0.41, p=0.53). (C) Same as in (B), but during resting before (Pre-run) and after (Post-run) running on the familiar track. Number of animals: $N$ = 5 (WT), 5 (RTT) for Pre-run and 7 (WT), 8 (RTT) for Post-run. Effect of genotype on occurrence rate ($F_{(1, 23)}$=0.15, p=0.70), duration ($F_{(1, 23)}$=0.49, p=0.49), amplitude ($F_{(1, 23)}$=0.12, p=0.73, and frequency ($F_{(1, 23)}$=0.06, p=0.81). The main effect of session on occurrence rate was no longer significant ($F_{(1, 26)}$=4.2, p=0.053), neither was on duration ($F_{(1, 26)}$=0.02, p=0.88), amplitude ($F_{(1, 26)}$=1.0, p=0.31), or frequency ($F_{(1, 26)}$=0.33, p=0.57). There was also no significant effect of the interaction between genotypes and resting sessions (Pre- and Post-run) on occurrence rate ($F_{(1, 21)}$=0.02, p=0.88), duration ($F_{(1, 21)}$=0, p=0.96), amplitude ($F_{(1, 21)}$=0.57, p=0.46), or frequency ($F_{(1, 21)}$=0.14, p=0.71).
DOI: https://doi.org/10.7554/eLife.31451.008
The following figure supplements are available for figure 4:

**Figure supplement 1.** Firing rates of CA1 cells in WT and RTT mice within ripples.
DOI: https://doi.org/10.7554/eLife.31451.009
**Figure supplement 2.** Run-active cells in RTT mice displayed enhanced activities within ripples.
DOI: https://doi.org/10.7554/eLife.31451.010
**Figure supplement 3.** Activities of run-inactive cells within ripples did not differ between WT and RTT mice.
DOI: https://doi.org/10.7554/eLife.31451.011

## RTT mice displayed normal contextual fear memory at short, but not at long, time scales

Our analysis so far indicates that place cell synchrony failed to increase after novel experience, likely due to baseline hypersynchrony, and place cell firing specificity was impaired on the familiar track. These results suggest that impaired memory consolidation contributes to memory deficits in RTT mice. We attempted to provide behavioral evidence for this hypothesis. Previous studies have shown that RTT mice are impaired in contextual fear memory when they are tested 3 or 24 hr after learning (*Samaco et al., 2013*; *Hao et al., 2015*). Here we examined the contextual fear memory of our RTT and WT mice at a shorter time scale, 20 min after learning, when presumably only minimal consolidation had occurred and memory should be relatively normal, according to our hypothesis. After exploring a novel fear-conditioning box and receiving two mild foot shocks, animals were returned to their home cages. We measured their freezing level 20 min later, at which time the animals were placed back to the conditioning box for 5 min. We found that the freezing level at 20 min was comparable between RTT and WT mice (*Figure 6A*). We also examined the freezing level of the mice at 1 and 24 hr later after the shocks and used the 24 hr freezing level as a measure of long-term memory, when presumably substantial ripple events and thus memory consolidation had occurred. ANOVA analysis revealed a significant effect of genotype and post-hoc Wilcoxon ranksum test showed that there was a significant difference at 24 hr, but not at 20 min, between WT and RTT mice. The memory of a subset of mice ($N$ = 12 WT, 14 RTT) was examined at all three time points. However, ANOVA analysis for these mice failed to identify a significant interaction of genotype by time ($F_{(2,72)}$ = 0.5; p=0.61), probably due to memory extinction of these animals that were tested repeatedly. Nevertheless, our data did show that the contextual memory at a short time scale (20 min) was relatively normal in RTT mice. This result, together with previous studies (*Samaco et al., 2013*; *Hao et al., 2015*) showing contextual fear memory deficits at longer time scales, is consistent with a memory consolidation impairment in RTT mice.

## Discussion

To understand how genetic deficiencies underlying Rett syndrome alter the function of memory circuits in vivo, we have examined CA1 place cells in freely behaving RTT and WT mice as they ran a linear track and as they rested in both a novel and familiar environment. We found that the firing specificity of place cells in RTT mice does not increase from the novel to the familiar environment as

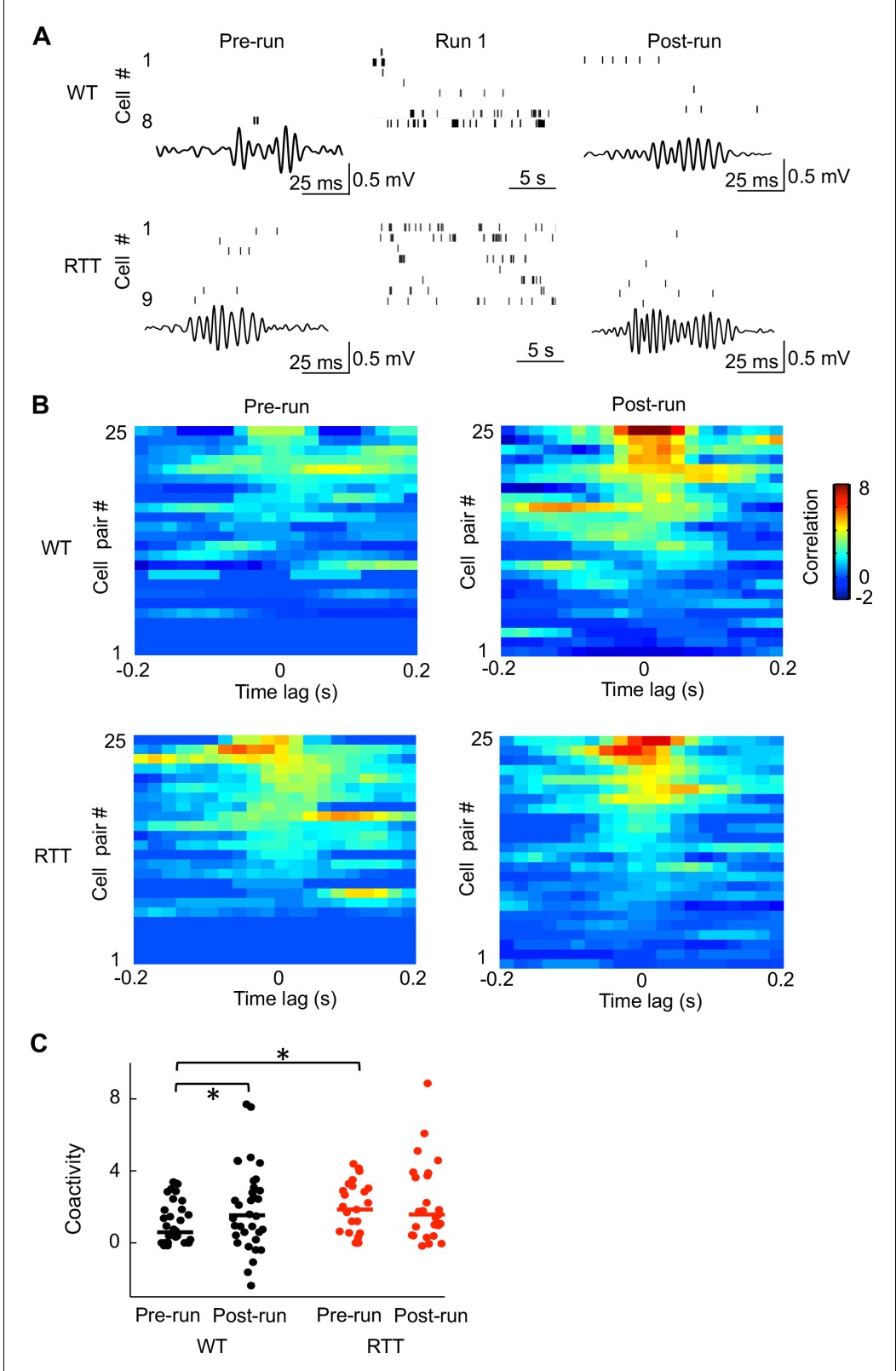

**Figure 5.** Run-active cells in RTT mice failed to increase their firing synchrony following running on the novel track. (**A**) Spiking activities of run-active cells in an example WT and RTT mouse within a ripple event in Pre-run and Post-run and within a running lap in Run 1. Each row represents a cell and each tick is a spike. Bottom trace: LFP filtered within the ripple band (100–250 Hz). (**B**) Cross-correlograms of pairs of run-active cells within ripples in Pre-run and Post-run. Pairs are ordered by their peak correlations. 25 pairs in RTT mice and a random subset of 25 pairs out of 34 in WT mice are

*Figure 5 continued on next page*

*Figure 5 continued*

plotted. All these pairs were recorded throughout all sessions of Pre-run, Run 1 and Post-run. (**C**) Coactivities of all run-active cell pairs in WT and RTT mice within ripple events in Pre-run and Post-run. Solid lines: median values. Number of pairs: *N* = 34 (WT), 25 (RTT). WT coactivity between Pre-run and Post-run: *p=0.044, paired Student's t-test; RTT: p=0.47. Pre-run coactivity between WT and RTT mice: *p=0.039, Wilcoxon ranksum test; Post-run: p=0.65.

DOI: https://doi.org/10.7554/eLife.31451.012

The following figure supplements are available for figure 5:

**Figure supplement 1.** CA1 cells displayed an overall baseline hypersynchrony in RTT mice.

DOI: https://doi.org/10.7554/eLife.31451.013

**Figure supplement 2.** Run-active cells in RTT mice displayed enhanced firing synchrony before (Pre-run) and after (Post-run) running on the familiar track.

DOI: https://doi.org/10.7554/eLife.31451.014

**Figure supplement 3.** Run-silent cells in RTT mice displayed enhanced firing synchrony in ripple events.

DOI: https://doi.org/10.7554/eLife.31451.015

much as that of place cells in WT mice (*Figure 6B*). In parallel to this dynamic change of place cells, contextual fear memory in RTT mice is normal at a short time scale (20 min), but is impaired at longer time scales (*Figure 6A*). These deficits point to a possible impairment in memory consolidation in RTT mice. Indeed, we found that run-active cells fail to increase their firing synchrony within ripples following the novel-track running (*Figure 6C*), likely due to an already high baseline synchrony (hypersynchrony) among CA1 cells in RTT mice. These results are consistent with our hypothesis that hippocampal spatial memory codes are abnormally refined with experience, due to impaired memory consolidation. Our finding may provide a neural circuit mechanism for memory deficits of Rett syndrome.

It is believed that spatial memories are initially encoded by a population of place cells active in a novel environment, driven by sensory input from the environment (*O'Keefe and Burgess, 1996*; *Knierim and Hamilton, 2011*). As novel environments become familiar, spatial memories are consolidated and their memory codes become refined, which reflect not only ongoing sensory input from the familiar environments, but also reflect the consolidated memories (*Frank et al., 2004*; *Cacucci et al., 2007*; *Lever et al., 2002*). Our data show that CA1 place cells in RTT mice contain a similar amount of spatial information as those in WT mice on the novel track, but form more place fields with relatively shorter field lengths. This suggests that, although certain aspects of the encoding are changed, ongoing sensory input is capable of driving place cells to encode novel spatial experience in RTT mice. Indeed, our data show that contextual fear memory in RTT mice is normal shortly after learning, presumably because place cells in RTT mice can initially encode the spatial context of a conditioning box. Our data also show that place cells in WT mice are more specific on the familiar track with narrower place fields and increased peak firing rates, compared to those on the novel track. In contrast, place cells in RTT mice increase their spatial specificity from the novel to the familiar track much less than those in WT mice. Although the smaller number of recorded cells on the novel track might not produce sufficient statistical power to identify possible significant differences between WT and RTT cells in parameters such as firing rate or ripple participation rate, our key finding of impaired increase in spatial information of RTT cells in the familiar environment is robust even with downsampled cells. Consistent with this observation, RTT mice are impaired in contextual fear memories tested 3 or 24 hr after learning in previous studies (*Samaco et al., 2013*; *Hao et al., 2015*). Therefore, abnormal spatial memory code refinement may underlie behavioral deficits in RTT mice.

It is known that ripple evens are important in memory processing (*Girardeau et al., 2009*; *Ego-Stengel and Wilson, 2010*; *Wu et al., 2017*). Within ripple events during Pre-run prior to a novel spatial experience, our data show an enhanced correlation among place cells in RTT mice. This hypersynchrony may be related to the greater number of place fields or shorter field size seen on the novel track, but appears not sufficient to impact overall spatial information or the behavioral performance. Importantly, following the running on the novel track, run-active cells in RTT mice fail to increase their synchrony within ripples. This is in contrast to the known phenomenon that place cells active in a novel experience normally tend to fire together within ripples after the experience, as shown in previous studies (*Wilson and McNaughton, 1994*; *Ji and Wilson, 2007*) and in our data

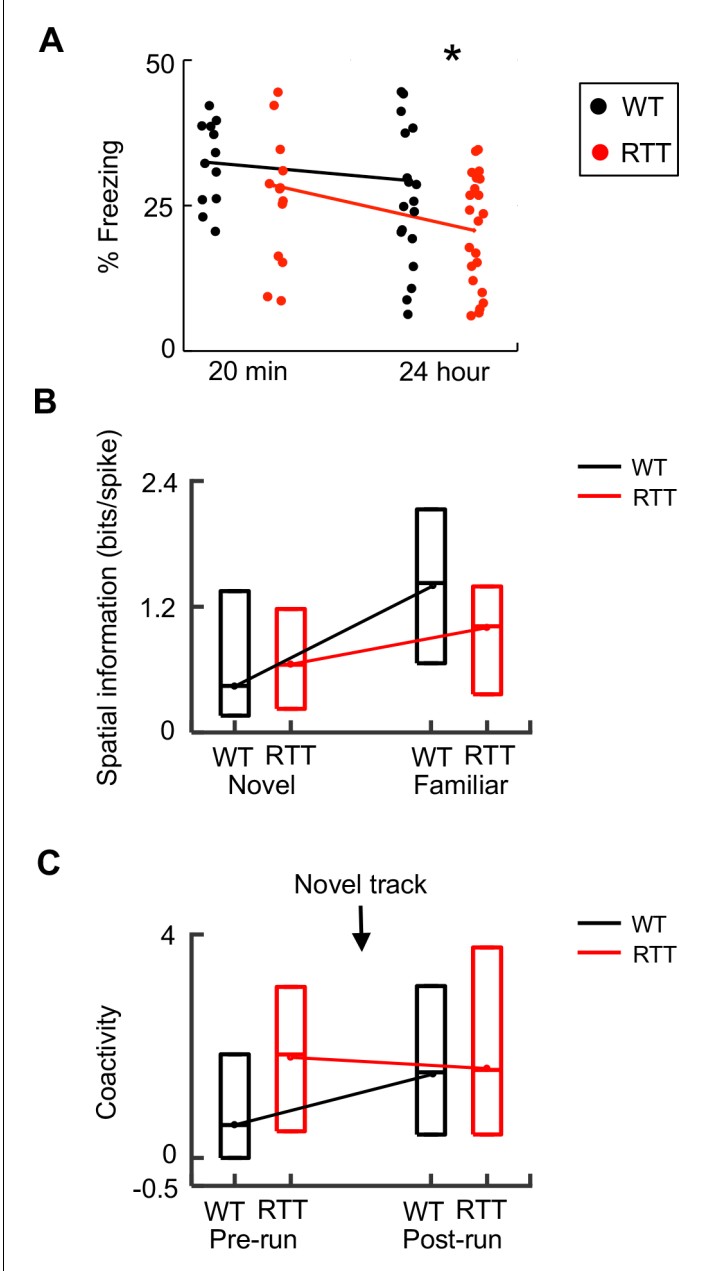

**Figure 6.** Dynamics of memory deficits paralleled those of neurophysiological deficits. (**A**) Change in memory deficit from short- to long-term memory. Percentages of freezing time in a fear conditioning box, tested 20 min and 24 hr after WT and RTT animals were foot-shocked in the box, are plotted. Each dot represents an animal. Lines are median values. Two-way ANOVA for the effect of genotype: $F(1,66) = 4.2$; p=0.044. Post-hoc Wilcoxon ranksum test at 20 min: p=0.78, $N$ = 12 WT, 14 RTT; at 24 hr: *p=0.034, number of animals: $N$ = 20 WT, 22 RTT. (**B**) Change in spatial information from novel to familiar track in WT and RTT mice. Data are shown as median and [25–75%] range values (same below, see *Figure 2*). (**C**) Change in coactivity of run-active cells before (Pre-run) and after (Post-run) running on a novel track in WT and RTT mice (see *Figure 5*).
DOI: https://doi.org/10.7554/eLife.31451.016

from WT mice. This experience-dependent increase in firing synchrony is hypothesized to be a neural mechanism for memory consolidation, because it likely results from synaptic plasticity during the novel experience and leads to further experience-specific synaptic plasticity for memory consolidation (*Buzsáki, 1989*; *Wilson and McNaughton, 1994*), via precise 'replay' of the activity patterns

associated with the experience (*Skaggs and McNaughton, 1996*; *Lee and Wilson, 2002*). Our finding that this experience-dependent increase in synchrony is reduced in RTT mice suggests an impairment in memory consolidation in these animals. In addition, we also found that place fields are relatively unstable between two running sessions of the novel track, providing further evidence that spatial memory codes fail to consolidate following the first running session. Given that there is hypersynchrony during Pre-run in RTT mice, it is likely that this baseline hypersynchrony occludes further experience-dependent increases in firing synchrony and thus prevents the experience-specific synaptic plasticity underlying memory consolidation, possibly due to saturation or other homeostatic mechanisms. Our previous study reported the hypersynchrony in hippocampal slices and when animals are quietly awake on an elevated resting platform (*Lu et al., 2016*). Our previous study also showed that the hypersynchrony, as well as memory deficits, in RTT mice can be rescued by deep-brain stimulation that promotes synaptic plasticity (*Hao et al., 2015*). Taken together with these previous findings, our data support a model that, although sensory input can drive the initial formation of spatial memory codes in novel environments, the baseline hypersynchrony leads to reduced experience-dependent increase in firing synchrony within ripples and consequently impairs memory consolidation and memory code refinement.

One issue that arises from our data is how to understand the change in firing synchrony within ripples in familiar environment. In WT mice, although there is still an increase in firing synchrony after running on a familiar track, the increase is less dramatic than that following running on a novel track. This is likely because experience-induced firing synchrony among run-active cells gets weaker when memories no longer need to be consolidated. In RTT mice, although hypersynchrony is still present in the rest session before running on the familiar track, the hypersynchrony among run-active cells becomes less prominent than that before running on the novel track. Also, running on the familiar track induces a larger increase in the firing synchrony than running on the novel track. It seems that run-active cells in RTT mice exhibit a change following the running on the familiar track similar to the cells in WT mice following the running on the novel track. One possibility is that this is caused by the repetitive overtraining experience on the familiar track, which may induce some long-term synaptic plasticity that 'breaks' the baseline hypersynchrony. The low baseline may now permit some experience-dependent increase in firing synchrony within ripples.

Our notion of abnormal memory consolidation caused by abnormal firing synchrony within ripples in RTT mice is consistent with a previous study on a mouse model of schizophrenia, which found that stronger correlations of place cell pairs within ripples lead to impaired reactivation of place cell patterns following an experience (*Suh et al., 2013*). However, in our study the abnormally high correlation occurs prior to any experience, and therefore the abnormal memory consolidation appears to be caused by a baseline hypersynchrony within ripples during resting. In addition, our idea that abnormal place field refinement in RTT mice is linked to abnormal neuronal activities within ripples is supported by a recent study that silencing place cells within ripples impairs refinement of their place fields (*Roux et al., 2017*). Finally, our finding of unstable place fields on the novel track in the face of hypersynchrony within ripples is consistent with several papers that study how manipulating neuronal activities within ripples affects place field stability. Silencing place cells within ripples leads to unstable place fields of these cells (*Roux et al., 2017*). Disrupting ripples altogether did not impact place field stability in one study (*Kovács et al., 2016*), but did reduce stability of a portion of place cells in another study (*van de Ven et al., 2016*). In any case, place cell silencing or ripple disruption is a much more severe alteration than the hypersynchrony in RTT mice, yet the hypersynchrony in RTT mice seems sufficient to impact place field stability.

Although we need to be cautious when interpreting data from animal studies, our study may provide insights into the cognitive deficits in human patients. The RTT mouse model shares the same genetic deficiencies that cause Rett syndrome and recapitulates key behavioral deficits in Rett patients, including impaired learning and memory (*Samaco et al., 2013*). It is likely that the in vivo neural circuit deficits identified here in RTT mice may also occur in human patients. Indeed, there is evidence that the increased synchrony described here is paralleled by human studies of Rett patients using fMRI or EEG recordings, which have shown that hypersynchronous neural activity at baseline is a hallmark of the disease (*Naidu et al., 2001*; *Garofalo et al., 1988*; *Glaze, 2005*). Our data suggest that baseline hypersynchrony in RTT mice reduces experience-dependent increase in firing synchrony for memory consolidation and consequently leads to impaired spatial memory codes. Our animal

study on RTT mice thus suggests that abnormal memory codes caused by impaired memory consolidation underlie the cognitive deficits in Rett syndrome.

# Materials and methods

## Key resources table

| Reagent type | Designation | Source or reference | Identifiers | Additional information |
|---|---|---|---|---|
| Gene (mouse) | *Mecp2* | DOI: 10.1038/85899 | RRID:MGI:3624717 | |
| Strain (mouse) | Wild-Type control | Source (public (Taconic Farms, Inc., Germantown, NY, USA)); Source (DOI: 10.1038/85899) | 129SvEv: RRID:MGI:5653381; FVB: RRID:IMSR_TAC:fvb | |
| Strain (mouse) | RTT | PMID: 11242117; DOI: 10.1038/85899 | RRID:IMSR_JAX:003890 | |
| Software | Neuralynx Cheetah | https://neuralynx.com/software/cheetah | | |
| Software | MatLab | https://www.mathworks.com/products/matlab.html | | |
| Software | xclust2 | https://github.com/wilsonlab/mwsoft64/tree/master/src/xclust | | |
| Software | DataManager | https://github.com/DaoyunJiLab/DataManager | | Copy archived at https://github.com/elifesciences-publications/DataManager |

## Animals

A total of 9 wild-type (WT) and 10 Rett (RTT) mice were used for electrophysiological recordings. Among them, 8 WT and 9 RTT mice yielded electrophysiological data during at least one behavioral procedure (see *Table 1* for details) and were used for the subsequent fear conditioning task. Given the large amount of effort in tetrode recording of each individual animal, to reduce unnecessary effort, the animals were genotyped before the recording experiment and the genotype was not blind to the experimenter. However, to ensure that WT and RTT animals followed the same experimental procedure, we always conducted experiments on a pair of WT and RTT mice at the same time. The number of animals was decided based on existing literature, where a typical tetrode recording experiment uses 3–10 animals per group. Additionally, two non-recorded cohorts totaling 11 WT and 12 RTT mice were used for fear conditioning only. F1 hybrid of FVB/N x 129S6/SvEv (FVB.129F1) animals were generated by mating female *Mecp2$^{+/-}$* mice which were maintained on a pure FVB/N background (RRID:IMSR_TAC:fvb) to male WT mice of a pure 129SvEv background (RRID:MGI:5653381). These mice were maintained on a 12 hr light:12 hr dark cycle, fed standard mouse chow and water *ad libitum* until the experiment began. The *Mecp2$^{tm1.1Bird}$* allele (RRID:IMSR_JAX:003890), lacking exons 3 and 4, was used in the *Mecp2$^{+/-}$* mice, and it generated a true *Mecp2*-null allele (*Guy et al., 2001*). Adult mice at ages of at least 3 months were used for electrophysiological recording and fear conditioning. Only female mice were used. Animals were housed 4 to 5 per cage until tetrode hyperdrives were implanted, then mice were single housed. All research and animal care procedures followed the recommendations in the 'Guide for the Care and Use of Laboratory Animals' of the National Institute of Health and were approved by the Baylor College of Medicine Institutional Animal Care and Use Committee.

## Surgery

Mice were implanted with a tetrode hyperdrive containing nine tetrodes (eight for recording, and one as a reference). Mice were anesthetized with 0.5–2% isoflurane and fixed using a stereotaxic device throughout the surgery. The implantation was made at 2.0 mm antereoposterior and 1.5 mm mediolateral from Bregma, targeting CA1 cells in the right hemisphere. The cannulae of the hyperdrive was affixed to the skull using dental cement and stainless steel anchoring screws. Supportive care of ketoprofen (5 mg/kg) and saline (1 ml) were injected prior the beginning of surgery.

## Tetrode recording

Tetrode recordings were made as previously described (*Cheng and Ji, 2013*; *Ciupek et al., 2015*). In 2–4 weeks following the surgery, the tetrodes were lowered slowly into the CA1 pyramidal cell

layer. Identification of the CA1 pyramidal cell layer was made by visual inspection of the spiking activity and sharp-wave ripples in the local field potentials (LFPs) while the animals were at rest. A Neuralynx Digital Lynx system was used to record both LFPs and neuronal spiking activity. Recordings were made while animals ran on a rectangular shaped linear track, or were on an elevated platform (*Figure 1B*). Spiking activity was identified with a pre-set threshold of 50–70 µV, sampled at 32 kHz, and was band-pass filtered from 0.6 to 6 kHz. LFPs were band-pass filtered from 0.1 to 1 kHz and sampled at 2 kHz. Animals' positions were monitored by two color diodes mounted over the tetrode hyperdrive and sampled at 33 Hz. Data was analyzed offline.

## Behavioral apparatuses and tasks

### Track-running and resting tasks

Animals (*N* = 10 RTT, 9 WT mice) were food deprived to ≥85% of their *ad libitum* weight ~2 weeks after recovery from surgery. They were first trained to run back and forth (two trajectories) on a familiar, rectangular-shaped track of ~2 m length for a food reward (condensed milk) at each end of the track in a familiar room for at least 2 weeks. Recordings started when the mice were able to run consistently back and forth on the track. On each of the recording days, the animal had two track-running sessions (Run 1, Run 2), each lasting ~15 min (*Figure 1B*). Then, a subset of animals (*N* = 9 RTT, 7 WT mice) was also recorded while they ran on a novel track for two sessions in a novel room, following the same daily procedure as before. The novel track had a similar shape to the familiar one, but with different tactile and visual cues. The animals had never been exposed to the novel room or the novel track before the recording. Some of the animals were also recorded while they were resting for 10–15 min on an elevated platform elevated platform before (Pre-run) and after (Post-run) the first track session on either the familiar or the novel track. For the familiar track, 5 RTT and 5 WT mice were recorded during Pre-run and 8 RTT and 7 WT mice recorded during Post-run. For the novel track, 7 RTT and 5 WT mice were recorded in Pre-run and 9 RTT and 7 WT mice recorded in Post-run.

### Fear Conditioning

22 RTT and 20 WT mice were studied, including those implanted, which were tested in fear conditioning after all recordings were completed. The animals explored a novel fear-conditioning box for 2 min, then a 30 s tone (80 dB, 5 kHz) was played with a coinciding mild footshock (2 s, 0.7 mA) at the end of the tone. After 1 min, the tone/footshock pairing was repeated. The animals stayed in the box for one more minute before returned to their home cages. To test contextual fear memory, 1 hr and 24 hr later, the animals were placed back to the conditioning box for 5 min and their freezing behavior was measured. A subset of animals (12 WT, 14 RTT) were also tested at an additional time point of 20 min after conditioning, prior to the testing at other two time points. Data at 20 min and 24 hr were plotted to assess contextual memories before and after memory consolidation.

## Histology

After all recordings, each recorded animal was euthanized using a pentobarbital overdose (50 mg/kg). Current of 30 µA was passed through each tetrode for 10 s to generate a small lesion at each recording site. The brain was then dissected out, fixed in 10% formaldehyde solution for at least 24 hr, cryoprotected in 30% sucrose, and then sectioned at 50 µm thickness. Slices were placed on slides and dried. They were then stained using 0.2% Cresyl violet and cover-slipped. Recording sites were then identified by matching lesion sites with tetrode depths and relative positions. Only tetrodes in the CA1 pyramidal cell layer were used for analysis.

## Data analysis

For all recording days, single units were sorted offline using a manual clustering software (https://github.com/wilsonlab/mwsoft64/tree/master/src/xclust) (*Wilson, 2008*). Only well-sorted clusters (with isolation distance >9.5) were included in the analysis. Although animals were recorded across several days on the familiar track, only one day's data was used in the analysis for these behavioral conditions, to avoid possible repeated sampling of same cells. Only putative pyramidal cells (average firing rate ≤7 Hz) were included in the analysis. Behavioral and neural spike data were analyzed using a custom Matlab package DataManager (https://github.com/DaoyunJiLab/DataManager)

(*Ji, 2018*; copy archived at https://github.com/elifesciences-publications/DataManager). Results are presented with median and quartile [25–75%] values unless otherwise indicated. Statistical comparisons were completed using the non-parametric Wilcoxon ranksum test or ANOVA, unless otherwise specified.

## Firing rate curve and place field quantification

We identified every time period (lap) an animal traveled through one of two trajectories on a track. Each trajectory was separately linearized and divided into 2 cm spatial bins. Run-active cells, defined as those with average rate >0.5 Hz on at least one trajectory in one track-running session, were analyzed on each trajectory they were active on. For each cell on a trajectory, we computed its firing rate ($x_i$) within each spatial bin ($i$), as the total number of spikes divided by the total occupancy time ($t_i$) across all laps on that trajectory in a session. The cell's firing rate curve (firing rate at each bin: [$x_1$, $x_2$, ..., $x_N$]) was smoothed by a Gaussian kernel with a sigma of 2 bins. For this analysis, we excluded the spatial bins at the reward site and those stopping periods, when animal did not move (speed <3 cm/s for more than 1 s). Spatial information was calculated from a rate curve as previously described (*Sirota et al., 2003*). Spatial information was computed on each active trajectory and averaged between the two running sessions (Run 1, Run 2). Stability was the Pearson correlation between a cell's rate curves on the same active trajectory across two running sessions (*Ciupek et al., 2015*). Place fields were defined as peaks of a firing rate curve with a peak rate ≥1 Hz. A threshold of 10% of peak rate was used to detect place field boundaries (*Mehta et al., 2002*). Place fields with a gap of ≤6 cm were combined into one.

## Downsampling analysis

For downsampling analysis, the 4 groups of cells, WT and RTT cells active on the familiar and novel tracks, were equalized to have the same number of cells, which was the number of RTT cells on the novel track (31). To do so, 31 cells were randomly chosen among each of the other three groups. This random downsampling was repeated 200 iterations. Since the downsampling largely did not change the cells on the novel track, it mostly affected the results involving cells on the familiar track. Therefore, we subjected a key variable, spatial information, to the downsampling analysis. For each iteration, the spatial information of the chosen cells was statistically compared between WT and RTT cells or between the novel and familiar tracks, using the Wilcoxon ranksum test. For each comparison, the distribution of resulting *P* values was plotted and the proportion of iterations that produced a significant *P* value (<0.05) was reported (*Figure 2—figure supplement 1*).

## Theta phase precession

LFPs were filtered through the theta frequency range of 6–10 Hz. For all spikes within a place field, we computed the regression between their phases relative to the filtered theta LFPs and their firing locations. Two types of regression were used. The first was the optimal linear regression as in the original phase precession paper (*O'Keefe and Recce, 1993*), where a linear regression was computed with spike phases shifted by every 1° among a 360° range and the regression with the best fit was used. The second was the circular-linear regression between the circular phase values and spike locations, with the formula given in a previous study (*Ravassard et al., 2013*).

## Ripple event detection and quantification

Individual ripple events were detected via CA1 LFPs recorded while mice were resting before (Pre-run) and after (Post-run) track-running, as in previous studies (*Csicsvari et al., 2000*; *Lee and Wilson, 2002*). A LFP trace was first filtered through the ripple band (100–250 Hz) and its standard deviation (std) was computed. A (peak) threshold of 6 stds was used to identify potential events. Start and end times of ripple events were determined by a threshold of 2.5 stds. Neighboring events with a gap of <30 ms were combined into single events. We defined those identified events with durations between 30 to 500 ms as ripple events. Ripple parameters, including occurrence rate (number of ripples per second), amplitude (the absolute value of the most negative trough), duration, frequency, were quantified for each ripple event and averaged across all ripple events in an animal for each rest session. For each putative pyramidal cell, we quantified its ripple participation rate as the ratio of number of ripples when the cell fired at least one spike over the total number of ripples in a session.

The numbers of spikes within all ripple events in a session were also averaged and reported as number of spikes per ripple event.

## Pairwise cross-correlation and coactivity

We computed pairwise spike cross-correlation within ripple events, similarly to previous studies (*Sirota et al., 2003*; *Ciupek et al., 2015*). For each ripple event in a session, we identified its start and end times as described above and defined a time period of 150 ms before the start time and 150 ms after the end time. For each pair of neurons, we computed a spike-count cross-correlation correlogram for spikes within these ripple time periods (*Terada et al., 2017*; *Diba et al., 2014*), with a bin size of 20 ms, which was then smoothed by 5-bin nearest neighbor averaging. The spike-count cross-correlation was then subtracted by the average number of spikes among the baseline time lags ([−220–180] ms and [180 220] ms). Finally, the cross-correlograms were re-scaled by the square root of the number of spikes expected from two random Poisson spike trains with identical firing rates (*Sirota et al., 2003*; *Ciupek et al., 2015*). The normalization renders the cross-correlation insensitive to firing rates of the two neurons. For a pair of neurons, their coactivity within ripples events was defined as the average of the normalized cross-correlogram values at lag bins within [−50 50] ms (*Wilson and McNaughton, 1994*).

## Acknowledgements

This work was supported by the W M Keck Foundation (HYZ, DJ), NIH/NINDS grant 5R01NS057819 (HYZ), Simons Foundation (DJ), and partly by the Neuroconnectivity Core and Neurobehavior Core of IDDRC at Baylor College of Medicine (1U54HD083092). The content is solely the responsibility of the authors and does not necessarily represent the official views of the Eunice Kennedy Shriver National Institute of Child Health and Human Development or the National Institutes of Health. HYZ is an investigator with the Howard Hughes Medical Institute.

## Additional information

### Competing interests

Huda Y Zoghbi: Senior editor, *eLife*. The other authors declare that no competing interests exist.

### Funding

| Funder | Grant reference number | Author |
| --- | --- | --- |
| W. M. Keck Foundation | | Huda Y Zoghbi<br>Daoyun Ji |
| National Institutes of Health | 5R01NS057819 | Huda Y Zoghbi |
| Howard Hughes Medical Institute | | Huda Y Zoghbi |
| Intellectual and Developmental Disabilities Research Center | 1U54HD083092 | Huda Y Zoghbi |
| Simons Foundation | 273886 | Daoyun Ji |

The funders had no role in study design, data collection and interpretation, or the decision to submit the work for publication.

### Author contributions

Sara E Kee, Conceptualization, Data curation, Formal analysis, Methodology, Writing—original draft, Writing—review and editing; Xiang Mou, Software, Formal analysis, Writing - review and editing; Huda Y Zoghbi, Conceptualization, Supervision, Funding acquisition, Methodology, Project administration, Writing - original draft, Writing—review and editing; Daoyun Ji, Conceptualization, Supervision, Funding acquisition, Methodology, Project administration, Wriitng - original draft, Writing—review and editing

## Author ORCIDs
Xiang Mou (ID) https://orcid.org/0000-0002-8579-7316
Huda Y Zoghbi (ID) https://orcid.org/0000-0002-0700-3349
Daoyun Ji (ID) http://orcid.org/0000-0003-4115-5888

## Ethics
Animal experimentation: All research and animal care procedures followed the recommendations in the Guide for the Care and Use of Laboratory Animals of the National Institute of Health and were approved by the Baylor College of Medicine Institutional Animal Care and Use Committee (protocol #AN-5134).

## Decision letter and Author response
Decision letter https://doi.org/10.7554/eLife.31451.021
Author response https://doi.org/10.7554/eLife.31451.022

## Additional files
### Supplementary files
• Transparent reporting form
DOI: https://doi.org/10.7554/eLife.31451.017

### Data availability
Data generated and analyzed for this study have been deposited to Dryad.

The following dataset was generated:

| Author(s) | Year | Dataset title | Dataset URL | Database, license, and accessibility information |
| --- | --- | --- | --- | --- |
| Sara E Kee, Xiang Mou, Huda Y Zoghbi, Daoyun Ji | 2018 | Data from: Impaired spatial memory codes in a mouse model of Rett syndrome | http://dx.doi.org/10.5061/dryad.h752n4q | Available at Dryad Digital Repository under a CC0 Public Domain Dedication |

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
