## [Decision Letter]

Thank you for submitting your article "Spatial memory codes are unrefined in a mouse model of Rett syndrome" for consideration by *eLife*. Your article has been reviewed by Timothy Behrens as the Reviewing Editor and Senior Editor, and two reviewers. One of the reviewers, Fabian Kloosterman, has agreed to reveal his identity.

The reviewers have discussed the reviews with one another and the Reviewing Editor has drafted this decision to help you prepare a revised submission.

We have had a very clear discussion of the manuscript, the upshot of which is that the reviewers both find the manuscript interesting, but both have reservations that might be addressed with additional analyses, which are detailed in the reviews below.

There is one point that I should mention that emerged from the discussion. You will note that reviewer #2 has a major concern that would require extra data. This is that in many analyses on the novel track there is a paucity of cells and many of the cells in key analyses are from a single animal. This is noted in R2 point 1. In the discussion, it was agreed that this was a weakness, but it was also agreed that the situation is substantially better in the familiar track. The reviewers remained firm that they would like to see more data to support the claims as currently made in the manuscript, and they thought it possible that you already have the data available. However, if this not possible, the reviewers noted that they were open to a toning down of the interpretation of the data from the novel track, and a stronger focus on the data from the familiar track.

Summary:

Kee et al., use female Mecp2 +/- mice, a transgenic model of Rett syndrome, to examine changes in place coding and synchrony of CA1 pyramidal cells. This work follows published data from some of the same authors which identified hypersynchrony in the CA1 of these model mice. They report that on novel tracks CA1 place cells are indistinguishable between genotypes, however in familiar environments Mecp2+/- place cells showed higher firing rates, broader place fields and reduced spatial information compared to those in control mice. The authors then go on to characterize pyramidal cell activity during sharp-wave ripples and attribute the changes to a lack of experience dependent increases in synchrony between exposures, with wildtypes showing an increase but Mecp2+/- mice having an elevated baseline synchrony that fails to increase further following experience. Finally, they use fear conditioning to demonstrate that these changes while important for long term memory are dispensable for short term retrieval of contextual fear memory.

*Reviewer #1:*

Kee et al., use female Mecp2 +/- mice, a transgenic model of Rett syndrome, to examine changes in place coding and synchrony of CA1 pyramidal cells. This work follows published data from some of the same authors which identified hypersynchrony in the CA1 of these model mice. They report that on novel tracks CA1 place cells are indistinguishable between genotypes, however in familiar environments Mecp2+/- place cells showed higher firing rates, broader place fields and reduced spatial information compared to those in control mice. The authors then go on to characterize pyramidal cell activity during sharp-wave ripples and attribute the changes to a lack of experience dependent increases in synchrony between exposures, with wildtypes showing an increase but Mecp2+/- mice having an elevated baseline synchrony that fails to increase further following experience. Finally, they use fear conditioning to demonstrate that these changes while important for long term memory are dispensable for short term retrieval of contextual fear memory.

While the data and their interpretations are interesting and are consistent with existing publications in the field, I have several issues with the manuscript and underlying data set that would preclude publication at this stage. I would suggest the authors include significantly more data and use a more appropriate method for the cell-pair analysis, detailed below.

Essential revisions:

1) My primary concern with these data is the small sample size and the outsized contribution of individual animals to the overall data. For example, the authors only analyze 29 place cells in the RTT mice on the novel track and 66 on the familiar. These low overall numbers make the analyses of the data very susceptible to outliers. Further, while I appreciate the breakdown of data in Table 1, it reveals that in several cases one or two mice of a given genotype contribute the vast majority of data to a given analyses. 2 wildtype mice contribute nearly 70% of data to novel track place field analyses and 85% of the pairwise analyses data. Similarly, a single RTT mouse contributes 76% of the pairs analyzed during slow-wave sleep. Given that all these measures vary with the behavior, experience and state of a single mouse it is crucial that a more distributed and representative data set be collected. I would urge the authors to add additional animals to their dataset to more robustly prove their findings. In line with this, I think that several of their animals need to be excluded completely from their analyses. Recording from CA1 with eight tetrodes and having a total yield of 5 or less place cells suggests that something has gone very wrong, and the data are most likely not reliable. I would ask the authors to add several good mice to their dataset (for example, comparable to the first two wildtype mice in Table 1) for both genotypes and confirm that their results stand or, are indeed strengthened by additional data.

2) The main focus of the paper is on place cell activity during exploration and their pre and post activity during ripples in the rest sessions. I therefore don't really understand the inclusion of non-track active putative pyramidal cell data and think this would only act to confuse the reader.

3) I am confused with the place cell analyses as presented in the manuscript and believe that several additional restrictions in the analysis may produce more robust results. Subsection “Less specific place fields in the familiar, but not novel, environment”, do the authors think that a place cell having an average of 4.9 fields (or 4.6 for wildtypes) is consistent with place cell literature? Obviously in that population (with a mean of 4.9) I would expect some would have close to 10 fields, which on an approximately 1.5m track is not believable. I understand and respect that there is variability across labs on how to define this measure and what is important is the between group comparisons, nonetheless this should be addressed.

4) Continuing from point 3, several key pieces of information are missing from the methods section which may clarify these data. First, the authors should state the linearized length of the tracks used. Next, there is no mention of any cluster quality metrics being calculated and cells excluded if they do/don't exceed a threshold such as: interspike interval, minimum spike number, complex spike index and isolation distance. Likewise, I see no mention of a lower velocity threshold for spike data, while running the linear track something that is routinely done to avoid false fields generated by discharge during rest/ripple periods on the track. The authors should check the number of spikes contributing to each 'field' and disregard any field containing less than 50 spikes or a similar sensible threshold. The authors may also want to increase their rather low 1 Hz threshold for initial place cell detection, to further avoid spurious place field detection. I would expect the mean number of place fields per cell to reflect the example shown in figure 3 A and C where most cells have 1, or 2 fields at most. As a result, the statement in Subsection “Less specific place fields in the familiar, but not novel, environment” about broader place fields in RTT cells should be re-evaluated following the changes outlined to cluster quality and place field detection used in the study.

5) A fundamental claim of this work is that there is not an experience-dependent increase in pairwise synchrony in the RTT mice following exploration of a novel track. Previous work however has shown this increase is contingent on co-activity of those place cells during running and/or a function of the physical distance between the centers of their place fields (Wilson and McNaughton 1994, Karlsson and Frank 2009, Suh et al., 2013). The conclusions of this paper as currently presented would require a similar approach to the analyses. The sample size and data quality issues would probably prevent this, thus without further data the conclusions would need to be softened and qualified.

6) I am confused by the authors approach to calculating correlations between spike trains. From the Materials and methods section and their references I understand they are applying a normalization step to correct for differences in firing rate, however in their previous work employing this method (Ciupek et al., 2015) and the reference they provide (Sirota et al., 2003) the resulting correlations were all positive values. Intuitively, cross correlating spike counts shouldn't give you any negative numbers, as the minimum number of spikes you can have is zero. In Figure 5—figure supplement 1B in this paper in the pretrack condition there is a large negative component evident- how is this to be interpreted? Moreover, in subsection “Lack of experience-dependent increase in place cell synchrony within ripples” the authors reference this figure as evidence that RTT cells show hypersynchrony in the pre-track condition compared to WT cells, however the medians are identical, both at zero.

7) While it appears the cross-correlation results (Figure 5B) is highly smoothed, possibly along both axes. This needs to be reported in the legend and/or methods, as smoothing along the y-axis should be avoided. Further regarding Figure 5B, subsection “Lack of experience-dependent increase in place cell synchrony within ripples”,the increased correlation at time zero in WT mice post-track is not at all clear from the plot, nor is the increased synchrony in RTT pre-track cell pairs compared to controls.

8) Regarding the correlation methods. The Diba et al., 2014 (Figure 2) and Terada et al., 2017 (Figure 5B right column) papers show what I would expect spike pair CCGs to look like for hippocampal neurons, consisting of positive values. The authors should consider replicating the findings with one of these accepted methods to demonstrate differences in synchrony during ripples.

9) I am also concerned that there are different numbers of mice contributing to the pre and post track session data. The changes the authors report between these periods in terms of CA1 firing rates and pairwise correlation should be shown to exist in the subset of mice recorded in both sessions.

10) I am puzzled by the difference in pretrack synchrony in the RTT mice between the novel and familiar recording session (Figure 5C and Figure 5—figure supplement 1B) The axes on these 2 figures, which could be thought of as equivalent baseline recording sessions- differ by a factor of 4. Again, the most reasonable explanation I can think of for this discrepancy is the sparseness of the data set.

11) Regarding the behavior – as written in the methods section, implies all mice were tested at 20 minutes and 24 hours, however the n is quite different in Figure 7A. Does the behavioral phenotype hold up if the analyses is restricted to the subset of mice tested at both time points and analyzed with a 2-way repeated measure ANOVA?

*Reviewer #2:*

In this manuscript, the authors characterize the activity of hippocampal place cells in a mouse model of Rett Syndrome. The main finding is that experience-dependent changes in the spatial firing properties of hippocampal place cells are diminished in the Rett (RTT) mouse model, a phenotype that is accompanied by hypersynchronous coactivation of hippocampal cells during ripple events that have been implicated in offline memory consolidation.

Overall, the results are interesting and – at the surface – the distinct findings reported in this study, i.e. higher firing rates (on track and during ripple events) in RTT mice, absence of spatial tuning refinement in RTT mice, increased baseline coactivity during ripple events without an experience-dependent increase, and abnormal contextual fear memory at 24 hours, support the conclusion that hypersynchronous hippocampal activity interferes with memory consolidation in this mouse model.

Still, I have a number of questions and some major concerns regarding the link between the different findings, the depth of the analyses that were performed and a few inconsistencies between the reported results and the data visualization. I believe most major concerns can be properly addressed with additional data analyses and clarifications in the text.

1) The authors report increased average firing rate (on the track and in ripple events) as well as hypersynchrony among CA1 cells during ripple events in RTT mice. How are these two phenomena related? Are cells active in more ripple events, or are cells emitting more spikes in the same number of ripple events? According to Figure 2, the firing rate changes during ripple events are observed in the population of all cells, but not (or much less so) in the subset of "track-active" cells. Could the authors confirm that in particular the "track-inactive" cells have an enhanced participation in ripple events?

2) The authors report increased baseline coactivity in RTT mice, which also expresses itself in the pre-track period before the mice ran on the novel track. However, according to the methods, the mice were trained on the familiar track before being introduced to the novel track. Do hippocampal cell populations in RTT mice show the same level of orthogonalization between the familiar and novel tracks (i.e. global remapping) as in WT mice? If not, could this explain the increased coactivity of novel track cells prior to the experience? Relatedly, the increase in synchrony/coactivity could reflect a stereotyped subset of cells that is more active than other cells, but not specifically tied to any given spatial environment.

3) The authors conclude that the observed hypersynchrony in RTT mice "interferes" with memory consolidation, possibly because the higher baseline hypersynchrony during ripple events in RTT mice may "occlude" experience-dependent increases in firing synchrony (Discussion section). The authors should clarify what they mean by "interfere" and "occlude" – e.g. following the classic model that spike sequence reactivation induces synaptic plasticity, do they mean that the higher baseline coactivation leads to a saturation of connection strengths between participating cells? Alternatively, if "track-inactive" cells participate more in ripple events (see point 1), it is conceivable that the deficits in memory consolidation and spatial tuning refinements are due to "imprecise" spike sequence reactivation that includes cells unrelated to the original experience.

4) The authors suggest that "spatial memory encoding is normal in RTT mice" (Discussion section) mainly based on the lack of clear differences in coarse spatial tuning properties between WT and RTT cells when mice run on a novel track. However, apart from the possibility that stereotyped cell populations are active in RTT mice (see above), spatial memory encoding may also be abnormal when studied at a more detailed level. For example, the authors should address the possibility that lap-to-lap refinements of place cell firing within a single recording session and/or theta sequences (which share some similarities with reactivation sequences during ripple events) are affected.

5) There are discrepancies and possible visualization errors in the reporting of the coactivity/correlation data (subsection “Baseline hypersynchrony within ripples during rest and sleep”,Figure 5 and Figure 6). The text and legend of Figure 6 mention a significantly higher coactivity in RTT as compared to WT, but Figure 6A and the summary statistics in the figure legend do not support this statement. Possibly related to this: there appears to be an error in the correlation plots in Figure 5—figure supplement 1 and Figure 6—figure supplement 1, as several traces appear to be "clipped" to zero (even though the legend claims that there is a significant difference between WT and RTT). Please double check the data – it is currently difficult to assess the validity of the results.

6) In the fear conditioning experiment, the same mice were tested three times at 20 min, 1 hour and 24 hours after acquisition. This introduces the possibility that reduced freezing at 24 hours that was observed in the RTT mice is due to enhanced extinction, rather than impaired consolidation. One way the authors could address this is by testing a group of mice only at 24 hours.

7) One of the main results is that WT cells, but not RTT cells, refine their spatial tuning properties in the familiar track (as indicated a spatial information increase – figure 3D). However, the place field length of WT cells appears to be larger in familiar as compared to novel track (figure 3B vs 3D), without an apparent decrease in the number of fields (subsection “Less specific place fields in the familiar, but not novel, environment”). Could the authors explain these seemingly opposing changes? They should also extend the comparison between novel and familiar spatial tuning properties in subsection “Less specific place fields in the familiar, but not novel, environment” to include place field length and peak rate measures.

[Editors' note: further revisions were requested prior to acceptance, as described below.]

Thank you for submitting your article "Impaired spatial memory codes in a mouse model of Rett syndrome" for consideration by *eLife*. Your article has been reviewed by Timothy Behrens as the Reviewing Editor and Senior Editor, and two reviewers.. One of the reviewers, Fabian Kloosterman, has agreed to reveal this identity.

The reviewers agree that the manuscript is nearly ready but there are a few outstanding comments below.

*Reviewer #1:*

Overall, I find the manuscript of Kee et al. to be much improved and appreciate the additional data and analyses the authors have provided. I have a few remaining points:

1) While the current manuscript is improved, I do still suspect that some of the differences in the novel and familiar track data related to the small sample size on the novel track. For example, median firing rate of run-active neurons is 1.1 Hz in WT on both types of track and 1.8 Hz in RTT mice in both types of track, yet only significant on the familiar due to the much more robust sample size. A similar trend is obvious in the pre-run ripple participation data. I would be more comfortable if the authors briefly acknowledge this possibility in the discussion.

2) I am confused by Figure 2—figure supplement 1. In the text the authors state that down sampling the WT cells still resulted in significantly lower spatial information in the RTT cohort. First, it's unclear in the text and methods how this was done- was down sampling repeated many times as is the norm, then each individual sample compared to the RTT distribution to find if a significant difference was present? If so, how many iterations were run and what fraction were significantly different? Also, the graph in panel A shows the opposite effect of what the authors conclude, with the distribution of the RTT population in red shifted to the right of the WT- please clarify.

3) The lap-by-lap data in Figure 2—figure supplement 2 is a nice addition, however I am not convinced the authors applied the correct statistical measure. I understand this is non-parametric data, hence their choice of the Wilcoxon ranksum, however this is clearly a repeated comparison between the groups, hence I believe the α must be adjusted for the number of comparisons (8) when claiming significance.

4) I still am a bit troubled by the behavioral data. The authors now include a 2-way ANOVA but fail to find the significant interaction between genotype and time which their conclusion would require. This should be made clear and the conclusions adjusted appropriately.

*Reviewer #2:*

The authors have made substantial changes to the manuscript, most notably adding extra mice to increase to number of recorded cells and performing additional analyses. As a result, the manuscript has improved considerably.

As the authors acknowledge, the discrepancy in the number of cells between novel and familiar conditions remains, and their explanation is reasonable. More importantly, the authors address the imbalance by subsampling the groups with high cell number (and limiting the contribution of individual mice) and show that main effect on spatial information remains.

Overall, I am very positive about the revised manuscript. I have a few minor remarks and a number of comments about the statistical tests that are applied.

---

## [Author Response]

There is one point that I should mention that emerged from the discussion. You will note that reviewer #2 has a major concern that would require extra data. This is that in many analyses on the novel track there is a paucity of cells and many of the cells in key analyses are from a single animal. This is noted in R2 point 1. In the discussion, it was agreed that this was a weakness, but it was also agreed that the situation is substantially better in the familiar track. The reviewers remained firm that they would like to see more data to support the claims as currently made in the MS, and they thought it possible that you already have the data available. However, if this not possible, the reviewers noted that they were open to a toning down of the interpretation of the data from the novel track, and a stronger focus on the data from the familiar track.

We conducted additional experiments on 2 mutant (RTT) and 1 wildtype (WT) mice and recorded a total of 24 neurons on the novel track. However, we have also applied a more stringent criterion for cell selection, as one of the reviewers requested, and thus lost 11 cells on the novel track. In the end, we have added 13 cells on the novel track experiment (and lost 16 cells on the familiar track). The difference in number of cells between novel and familiar tracks is improved, but still present. We have further addressed this difference by resampling down and plotting data using similar number of cells (see details in response to point #1 below). We would like to point out that recording on the novel track was harder than that on the familiar track, because we only had one chance on the novel track per animal, whereas on the familiar track we recorded repeatedly for a few days and then analyzed the one day with most cells.

In addition, we did tone down the writing: We changed the title and changed the description of the place cells on the novel track as having normal spatial information, but with some parameters of place fields altered. We have also revised the Discussion section accordingly.

Reviewer #1:

[…] Essential revisions:1) My primary concern with these data is the small sample size and the outsized contribution of individual animals to the overall data. For example, the authors only analyze 29 place cells in the RTT mice on the novel track and 66 on the familiar. These low overall numbers make the analyses of the data very susceptible to outliers. Further, while I appreciate the breakdown of data in Table 1, it reveals that in several cases one or two mice of a given genotype contribute the vast majority of data to a given analyses. 2 wildtype mice contribute nearly 70% of data to novel track place field analyses and 85% of the pairwise analyses data. Similarly, a single RTT mouse contributes 76% of the pairs analyzed during slow-wave sleep. Given that all these measures vary with the behavior, experience and state of a single mouse it is crucial that a more distributed and representative data set be collected. I would urge the authors to add additional animals to their dataset to more robustly prove their findings. In line with this, I think that several of their animals need to be excluded completely from their analyses. Recording from CA1 with eight tetrodes and having a total yield of 5 or less place cells suggests that something has gone very wrong, and the data are most likely not reliable. I would ask the authors to add several good mice to their dataset (for example, comparable to the first two wildtype mice in table 1) for both genotypes and confirm that their results stand or, are indeed strengthened by additional data.

1) We conducted additional experiments on 2 mutant (RTT) and 1 wildtype (WT) mice and recorded a total of 24 neurons on the novel track. However, we have also applied a more stringent criterion for cell selection, as one of the reviewer requested, and thus lost 13 cells. The main conclusions remain the same.

2) The issue of some animals yielding more cells than others remains. The recorded number of neurons from an animal depends on how many tetrodes are placed exactly at the thin pyramidal layer of the CA1, which in many cases is accidental. This is harder on the novel track since we had only one chance (one day) for recording, while on the familiar track we recorded for several days and analyzed the one day with most cells. In our hands, the yield per tetrode in mice is lower than in rats.

3) We agree with the reviewer that the issue of over-representation is a concern. To address this, we have repeated a key result with down-sampling to limit the contribution of animal to less than 30% (Figure 2—figure supplement 1).

We have used more stringent criteria for cell selection including cluster quality measures and others as suggested below. However, we still include the cells from animals with low yields in our analyses. If the cells met the criteria for inclusion and the animals were healthy and performed normally, we believe we need to report these cells. As we stated above, the yield was largely accidental, and we could not identify any valid behavioral or procedural reasons for excluding these animals. Also, only a very small number of cells (~7) were included this way and their inclusion does not affect the results.

2) The main focus of the paper is on place cell activity during exploration and their pre and post activity during ripples in the rest sessions. I therefore don't really understand the inclusion of non-track active putative pyramidal cell data and think this would only act to confuse the reader.

The point is to show that the hypersynchrony in Pre-run is not just specific to later run-active cells, but present in all cells. We believe this is useful for readers to know. However, to follow the reviewer’s suggestion and to keep the paper tight, we have moved the data related this point to a supplementary figure (Figure 5—figure supplement 1).

In addition, we do agree with the reviewer that we need to focus on track running behavior.

We have removed the non-specific analysis on cell activity during slow-wave sleep (SWS). The cells recorded during SWS were removed from Table 1.

3) I am confused with the place cell analyses as presented in the manuscript and believe that several additional restrictions in the analysis may produce more robust results. Subsection “Less specific place fields in the familiar, but not novel, environment”, do the authors think that a place cell having an average of 4.9 fields (or 4.6 for wildtypes) is consistent with place cell literature? Obviously in that population (with a mean of 4.9) I would expect some would have close to 10 fields, which on an approximately 1.5m track is not believable. I understand and respect that there is variability across labs on how to define this measure and what is important is the between group comparisons, nonetheless this should be addressed.

We apologize for the confusion. Place fields were computed independently on each of the 2 trajectories across 2 sessions. The previous numbers of place fields were the total numbers of place fields per cell identified over the 4 trajectories across 2 sessions. We have now revised the number as the number of place fields per active trajectory. This number computed from the updated data after imposing additional restrictions (see below) is 1.2 – 2.0 per cell per active trajectory, comparable with the existing literature.

4) Continuing from point 3, several key pieces of information are missing from the methods section which may clarify these data. First, the authors should state the linearized length of the tracks used. Next, there is no mention of any cluster quality metrics being calculated and cells excluded if they do/don't exceed a threshold such as: interspike interval, minimum spike number, complex spike index and isolation distance. Likewise, I see no mention of a lower velocity threshold for spike data, while running the linear track something that is routinely done to avoid false fields generated by discharge during rest/ripple periods on the track. The authors should check the number of spikes contributing to each 'field' and disregard any field containing less than 50 spikes or a similar sensible threshold. The authors may also want to increase their rather low 1 Hz threshold for initial place cell detection, to further avoid spurious place field detection. I would expect the mean number of place fields per cell to reflect the example shown in figure 3 A and C where most cells have 1, or 2 fields at most. As a result, the statement in Subsection “Less specific place fields in the familiar, but not novel, environment” about broader place fields in RTT cells should be re-evaluated following the changes outlined to cluster quality and place field detection used in the study.

We are sorry for the missing information. We have now revised the method section to clarify the points the reviewer raised here.

1) We specifically stated the length of the tracks (~2 m).

2) We have now removed all the stopping time periods on the track for our place field analysis. This is a very important point and we thank the reviewer for this. This step changed some of our reported numbers in place field parameters, including place field length and place field stability, but the main conclusion of less spatial information on the familiar track, but not on the novel track, remains.

3) Regarding cluster quality measures, we added a threshold for isolation distance (> 9.5) for unit selection and a threshold for number of spikes in a session (> 100). We did not use interspike interval, because it offers little selection. Hippocampal units from putative pyramidal neurons almost always produce 0 or few spikes within the refractory spiking period, whether they were contaminated by other spikes or not, due to their sparse firing. We also think complex spike index is a measurement of the property of a sorted unit and is biased against cells with lower rates. We believe it may not be the best as criterion for data selection.

4) We clarified the criteria for place field detection: (i) First, the overall average firing rate of a cell on a trajectory > 0.5 Hz; (ii) Second, the peak firing rate of the smoothed (σ = 4 cm) firing rate curve >= 1 Hz, which is not low in our opinion given that the rates were smoothed; (iii) field length > 6 cm. These criteria essentially exclude unreliable fields with small number of spikes per running lap. These criteria yielded ~1.6 fields per trajectory and a median field size of ~35 cm on the familiar track, which are comparable to previous studies in other types of WT mice and even rats.

5) A fundamental claim of this work is that there is not an experience-dependent increase in pairwise synchrony in the RTT mice following exploration of a novel track. Previous work however has shown this increase is contingent on co-activity of those place cells during running and/or a function of the physical distance between the centers of their place fields (Wilson and McNaughton 1994, Karlsson and Frank 2009, Suh et al., 2013). The conclusions of this paper as currently presented would require a similar approach to the analyses. The sample size and data quality issues would probably prevent this, thus without further data the conclusions would need to be softened and qualified.

We have checked our data. Despite additional new experiments, indeed there were not sufficient pairs of cells with overlapping place fields for this analysis. However, we would like to point out that here we study the broader co-activity of cell pairs within ripples than the previous studies the reviewer mentioned. Our focus is to measure broadly to what extent cells active together within a track tend to be active together in a ripple (as in replay events, during which the first and last cells may have non-overlapping place fields on the track yet tend to fire together in ripple replays).

6) I am confused by the authors approach to calculating correlations between spike trains. From the Materials and methods section and their references I understand they are applying a normalization step to correct for differences in firing rate, however in their previous work employing this method (Ciupek et al., 2015) and the reference they provide (Sirota et al., 2003) the resulting correlations were all positive values. Intuitively, cross correlating spike counts shouldn't give you any negative numbers, as the minimum number of spikes you can have is zero. In Figure 5—figure supplement 1B in this paper in the pretrack condition there is a large negative component evident- how is this to be interpreted? Moreover, in subsection “Lack of experience-dependent increase in place cell synchrony within ripples” the authors reference this figure as evidence that RTT cells show hypersynchrony in the pre-track condition compared to WT cells, however the medians are identical, both at zero.

We thank both reviewers for raising this important point and we have modified the analysis (see additional details in point #8 below).

1) Our method applies a normalization step to the regular spike count crosscorrelation. The normalization is necessary, because we need to average and compare correlation (coactivity) values among groups of cell pairs, but spike counts greatly depend on firing rates (total spikes) of a pair of cells. In our normalization scheme (see response to point #8 below), 0 represents baseline chance-level correlation predicted from firing rates, and negative values mean fewer spikes found than the baseline level (anti-correlation). Anti-correlation occurs obviously in time bins when two cells with non-zero firing rates do not fire together.

2) Our previous paper (Ciupek et al., 2015) and the referred paper (Sirota et al., 2013, which used a similar but not identical normalization scheme) computed cross-correlations during SWS, which have much longer time periods and thus contains many more spikes. The large number of spikes renders correlation values in large time lags to fluctuate smoothly around the baseline value of 0 (but not exactly 0, with some small positive/negative values). The correlation values of smaller time lags are positive due to the synchronized firing in SWS. This is also consistent with the SWS correlation plots in our previous submission (no longer present in the revised manuscript in order to focus on runactive cells).

3) In this study, our Pre-run and Post-run were short resting sessions (~15 minutes) and contained a much smaller number of ripple events than SWS. Some cell pairs did not fire at all during the ripple events. More importantly, with small bin size (<10 ms), many pairs had no spikes within a given time bin. As a result, the correlation values, and therefore medial values, are dominated by zeros and small negative values. To address this issue, we have made 2 modifications. (i) We used a longer time bin size (20 ms) and (ii) we smoothed cross-correlation values by 5-point nearest neighbor averaging.

7) While it appears the cross-correlation results (Figure 5B) is highly smoothed, possibly along both axes. This needs to be reported in the legend and/or methods, as smoothing along the y-axis should be avoided. Further regarding Figure 5B, line 186, the increased correlation at time zero in WT mice post-track is not at all clear from the plot, nor is the increased synchrony in RTT pre-track cell pairs compared to controls.

We agree with the reviewer and now the y-axis smoothing has been removed. We have also replotted the data by ordering peak correlation values. The differences between WT and RTT pairs are now visually obvious.

8) Regarding the correlation methods. The Diba et al., 2014 (Figure 2) and Terada et al., 2017 (Figure 5B right column) papers show what I would expect spike pair CCGs to look like for hippocampal neurons, consisting of positive values. The authors should consider replicating the findings with one of these accepted methods to demonstrate differences in synchrony during ripples.

3) Our method consists of two steps. The first step is the same as in Diba et al., 2014 and Terada et al., 2017, which computes a spike count correlation value at each time bin for two spike trains. (We have now added the papers in the references). Because cell counts greatly depend on firing rates (total spikes) of a pair of cells, here we need to apply a second, normalization step in order to average and compare correlation values among groups of cell pairs with different firing rates.

This normalization step basically subtracts a baseline spike count computed at large time lags ([180 – 220] ms and [-220 -180] ms for within-ripple correlations) from the spike count values. Then, we re-scale the spike count by the square root of the chance-level expected spike count per bin, given the firing rates of the two cells. This method removes the effect of firing rates and allows comparison of correlation values of cell pairs with different firing rates, as long as the same bin size is used, and the pairs are recorded with similar time duration.

However, we did recognize that the small bin size we used caused too many zeros and negative values at many bins for sparsely firing cells in a short period of time. We have revised the bin size from the previous 5 ms to 20 ms and applied a 5-point average smoothing to the computed cross-correlation values.

9) I am also concerned that there are different numbers of mice contributing to the pre and post track session data. The changes the authors report between these periods in terms of CA1 firing rates and pairwise correlation should be shown to exist in the subset of mice recorded in both sessions.

We have now reanalyzed and mainly reported the coactivity data restricted to those cell pairs recorded in both Pre and Post. The key results remain similar (Figure 5). When we addressed this concern, we realized that some of the cells recorded on the track did not have a Pre-run or Post-run session. Therefore, we removed the number of pairs from Table 1, because it did not reflect the pair numbers in Pre-run or Post-run session. The number of pairs is reported in each of the relevant figures (Figure 5, Figure 5—figure supplement 1, Figure 5—figure supplement 2).

10) I am puzzled by the difference in pretrack synchrony in the RTT mice between the novel and familiar recording session (Figure 5c and Figure 5—figure supplement 1B) The axes on these 2 figures, which could be thought of as equivalent baseline recording sessions- differ by a factor of 4. Again, the most reasonable explanation I can think of for this discrepancy is the sparseness of the data set.

Part of the confusion is due to presenting median values for correlation values with a small bin size, which are dominated by zeros. We have modified the quantification and data presentation. The results are more consistent after using the revised methods.

11) Regarding the behavior- as written in the methods section, implies all mice were tested at 20 minutes and 24 hours, however the n is quite different in Figure 7A. Does the behavioral phenotype hold up if the analyses is restricted to the subset of mice tested at both time points and analyzed with a 2-way repeated measure ANOVA?

The reviewer is correct in pointing out that not all animals were tested at 20 minutes, but all tested at 24 hours (and at the 1 hour time point). We have now clarified the issue and have added a few more animals. We have now reported the 2-way ANOVA statistics.

When we restricted the analysis to a subset of animals that were tested at both 20minute and at 24-hour time points (12 WT, 14 RTT), the 2-way ANOVA showed a similar genotype effect, but the difference at 24-hours between WT and RTT mice did not reach the significant level, possibly due to extra fear extinction by testing at 3 time points. However, the response at 20-minute had no extinction issue and should be valid. At 24hour time points, several previous studies (Samaco et al., 2013; Hao et al., 2016) already show the contextual fear memory deficit and our data including all animals confirm this result. Given the time-consuming nature of breeding these animals, we believe we should report the results with all animals we had at this time.

Reviewer #2:

In this manuscript, the authors characterize the activity of hippocampal place cells in a mouse model of Rett Syndrome. The main finding is that experience-dependent changes in the spatial firing properties of hippocampal place cells are diminished in the Rett (RTT) mouse model, a phenotype that is accompanied by hypersynchronous coactivation of hippocampal cells during ripple events that have been implicated in offline memory consolidation.Overall, the results are interesting and – at the surface – the distinct findings reported in this study, i.e. higher firing rates (on track and during ripple events) in RTT mice, absence of spatial tuning refinement in RTT mice, increased baseline coactivity during ripple events without an experience-dependent increase, and abnormal contextual fear memory at 24 hours, support the conclusion that hypersynchronous hippocampal activity interferes with memory consolidation in this mouse model.Still, I have a number of questions and some major concerns regarding the link between the different findings, the depth of the analyses that were performed and a few inconsistencies between the reported results and the data visualization. I believe most major concerns can be properly addressed with additional data analyses and clarifications in the text.1) The authors report increased average firing rate (on the track and in ripple events) as well as hypersynchrony among CA1 cells during ripple events in RTT mice. How are these two phenomena related? Are cells active in more ripple events, or are cells emitting more spikes in the same number of ripple events? According to Figure 2, the firing rate changes during ripple events are observed in the population of all cells, but not (or much less so) in the subset of "track-active" cells. Could the authors confirm that in particular the "track-inactive" cells have an enhanced participation in ripple events?

We thank the reviewer for the suggestion. We have compared the number of spikes and participation rate of run-active and run-inactive cells in ripples between WT and RTT cells. The activity of run-inactive cells in ripples was similar in WT and RTT mice. The run-active cells tend to participate more in ripples, consistent with their tendency to fire more during ripples, especially during Post-run sessions.

2) The authors report increased baseline coactivity in RTT mice, which also expresses itself in the pre-track period before the mice ran on the novel track. However, according to the methods, the mice were trained on the familiar track before being introduced to the novel track. Do hippocampal cell populations in RTT mice show the same level of orthogonalization between the familiar and novel tracks (i.e. global remapping) as in WT mice? If not, could this explain the increased coactivity of novel track cells prior to the experience? Relatedly, the increase in synchrony/coactivity could reflect a stereotyped subset of cells that is more active than other cells, but not specifically tied to any given spatial environment.

This is a great point. Unfortunately, we did not track same cells in both the familiar and novel environments. The recordings on the familiar track and novel track were conducted on different days with only one type of environment each day. Since we had two running sessions on the same track (to monitor stability), we could not do this on both tracks because mice could only run two or three sessions per day. More sessions would saturate their food intake and they would not be motivated enough to complete all sessions.

Our focus here is on the hypersynchrony and its impact on consolidation and place cells. The issue of remapping is interesting but can be answered by future experiments.

3) The authors conclude that the observed hypersynchrony in RTT mice "interferes" with memory consolidation, possibly because the higher baseline hypersynchrony during ripple events in RTT mice may "occlude" experience-dependent increases in firing synchrony (Discussion section). The authors should clarify what they mean by "interfere" and "occlude" – e.g. following the classic model that spike sequence reactivation induces synaptic plasticity, do they mean that the higher baseline coactivation leads to a saturation of connection strengths between participating cells? Alternatively, if "track-inactive" cells participate more in ripple events (see point 1), it is conceivable that the deficits in memory consolidation and spatial tuning refinements are due to "imprecise" spike sequence reactivation that includes cells unrelated to the original experience.

Again, we thank the reviewer for this point. Our additional analysis found no difference in the participation rate of run-inactive cells in ripples, suggesting that the higher baseline coactivity of CA1 cells somehow makes the connections among the hippocampal cells (presumably CA3 cells) less plastic, which is supported by our previous study (Hao et al., 2016). This is consistent with the reviewer’s saturation hypothesis. We have added this point to the Discussion section.

4) The authors suggest that "spatial memory encoding is normal in RTT mice" (Discussion section) mainly based on the lack of clear differences in coarse spatial tuning properties between WT and RTT cells when mice run on a novel track. However, apart from the possibility that stereotyped cell populations are active in RTT mice (see above), spatial memory encoding may also be abnormal when studied at a more detailed level. For example, the authors should address the possibility that lap-to-lap refinements of place cell firing within a single recording session and/or theta sequences (which share some similarities with reactivation sequences during ripple events) are affected.

We appreciate the comment. Unfortunately, the small number of simultaneously recorded cells with active place fields on the same trajectory prevents us analyzing theta sequences. Nevertheless, we have added additional analyses on the lap-bylap changes in spatial information (Figure 2—figure supplement 2) and theta phase precession (Figure 3). While theta phase precession was largely normal, the lap-by-lap refinement revealed that cells in RTT started with lower spatial information than those in WT mice on the familiar track but started with similar spatial information on the novel track.

5) There are discrepancies and possible visualization errors in the reporting of the coactivity/correlation data (subsection “Baseline hypersynchrony within ripples during rest and sleep”,Figure 5 and Figure 6). The text and legend of Figure 6 mention a significantly higher coactivity in RTT as compared to WT, but Figure 6A and the summary statistics in the figure legend do not support this statement. Possibly related to this: there appears to be an error in the correlation plots in Figure 5—figure supplement 1 and Figure 6—figure supplement 1, as several traces appear to be "clipped" to zero (even though the legend claims that there is a significant difference between WT and RTT). Please double check the data – it is currently difficult to assess the validity of the results.Please see our response to points #6,8 of the reviewer #2. The problem was due to presenting median correlation values with small bin size, which are dominated by zeros. We have modified the bin size and data presentation in these figures and other relevant figures. The results are now visually obvious.6) In the fear conditioning experiment, the same mice were tested three times at 20 min, 1 hour and 24 hours after acquisition. This introduces the possibility that reduced freezing at 24 hours that was observed in the RTT mice is due to enhanced extinction, rather than impaired consolidation. One way the authors could address this is by testing a group of mice only at 24 hours.

This is a valid point. Our previous experiments (Samaco et al., 2013; Hao et al., 2016) have already shown that there is a significant impairment in contextual fear memory at 24 hours.

7) One of the main results is that WT cells, but not RTT cells, refine their spatial tuning properties in the familiar track (as indicated a spatial information increase – figure 3D). However, the place field length of WT cells appears to be larger in familiar as compared to novel track (figure 3B vs 3D), without an apparent decrease in the number of fields (subsection “Less specific place fields in the familiar, but not novel, environment”). Could the authors explain these seemingly opposing changes? They should also extend the comparison between novel and familiar spatial tuning properties in subsection “Less specific place fields in the familiar, but not novel, environment” to include place field length and peak rate measures.

We thank the reviewer for pointing this out. In our last submission, we did not exclude stopping time periods on the track, which produced many seemingly narrow place fields. We have made the correction, which have changed some of our conclusions regarding place field properties. Nevertheless, the main conclusion that spatial information was no different between WT and RTT mice on the novel track, but became so on the familiar track, remains. As the reviewer predicted, the place field length in WT mice on the familiar track was significantly shorter than that on the novel track.

We have also extended the comparison between novel and familiar place fields to include place field length and peak rate. Our result shows that the smaller spatial information in RTT mice on the familiar track was largely due to larger place field sizes.

[Editors' note: further revisions were requested prior to acceptance, as described below.]

The reviewers agree that the manuscript is nearly ready but there are a few outstanding comments below.

Reviewer #1:

Overall, I find the manuscript of Kee et al. to be much improved and appreciate the additional data and analyses the authors have provided. I have a few remaining points:1) While the current manuscript is improved, I do still suspect that some of the differences in the novel and familiar track data related to the small sample size on the novel track. For example, median firing rate of run-active neurons is 1.1 Hz in WT on both types of track and 1.8 Hz in RTT mice in both types of track, yet only significant on the familiar due to the much more robust sample size. A similar trend is obvious in the pre-run ripple participation data. I would be more comfortable if the authors briefly acknowledge this possibility in the discussion.

We agree with the reviewer. We have added a sentence in the Results section and one in the Discussion section.

2) I am confused by Figure 2—figure supplement 1. In the text the authors state that down sampling the WT cells still resulted in significantly lower spatial information in the RTT cohort. First, it's unclear in the text and methods how this was done- was down sampling repeated many times as is the norm, then each individual sample compared to the RTT distribution to find if a significant difference was present? If so, how many iterations were run and what fraction were significantly different? Also, the graph in panel A shows the opposite effect of what the authors conclude, with the distribution of the RTT population in red shifted to the right of the WT- please clarify.

(i) We have added a paragraph in the Materials and methods section on downsampling. (ii) We thank the reviewer for pointing out the discrepancy in Panel A of this figure. It was a mistake in assigning plot colors. We have made the correction.

(iii) We thank the reviewer for the comment, which made us realize single iteration of downsampling is not that meaningful. We have now provided distributions of significant level (*P* values) among 200 downsampling iterations in Figure 2—figure supplement 1 and added a paragraph in the result section. The new results show that about half of the iterations produced significant *P* values for comparing spatial information between WT and RTT cells in the familiar environment, as expected from reduced statistical power. However, about 98% of iterations produced significant *P* values for comparing spatial information of WT cells between novel and familiar tracks, while only 25% for comparing spatial information of RTT cells. The result supports our key conclusion that WT cells show strong experience-dependent refinement in spatial memory codes, but not RTT cells, which is the emphasis throughout the manuscript (e.g. the conclusion Figure 6). We have also revised the text to recognize the sample size issue as in the response to #1 above and toned down the difference between WT and RTT cells.

3) The lap-by-lap data in Figure 2—figure supplement 2 is a nice addition, however I am not convinced the authors applied the correct statistical measure. I understand this is non-parametric data, hence their choice of the Wilcoxon ranksum, however this is clearly a repeated comparison between the groups, hence I believe the α must be adjusted for the number of comparisons (8) when claiming significance.

We thank the reviewer for pointing this out. We have adjusted the significance level to 0.00625 and marked those laps only with *P* values lower than this level. Also, as suggested by reviewer #3, we combined data in the first 3 laps and those in the last 3 laps and did statistical comparisons for the groups separately with a significance level of 0.025.

4) I still am a bit troubled by the behavioral data. The authors now include a 2-way ANOVA but fail to find the significant interaction between genotype and time which their conclusion would require. This should be made clear and the conclusions adjusted appropriately.

Given that the deficit in contextual fear memory of RTT mice at 24 hours has been studied in multiple previous studies (e.g., Samaco et al., 2013; Hao et al., 2016), our purpose here is to show that the contextual memory at 20 minutes in RTT mice was relatively normal, which our data are valid to support. Since the memory in the mice for the 2-way ANOVA was tested at 3 time points, it is likely the weaker memory deficit at 24-hours in these mice was due to memory extinction. Therefore, we believe our behavioral data together with previous studies are consistent with our conclusion. Nevertheless, we agree with the reviewer that the negative 2-way ANOVA test for these animals need to be made clear and the conclusion needs to be adjusted. We have revised the text to do so and have toned down the conclusion accordingly.